# Seeking enlightenment of fluvial sediment pathways by OSL signal bleaching of river sediments and deltaic deposits

Elizabeth Chamberlain[1,2,3*] & Jakob Wallinga[3]

[1]Department of Earth and Environmental Sciences, Tulane University, New Orleans, LA, USA
[2]Department of Earth and Environmental Sciences, Vanderbilt University, Nashville, TN, USA
[3]Netherlands Centre for Luminescence Dating & Soil Geography and Landscape group, Wageningen University, Wageningen, The Netherlands

*Correspondence to*: Elizabeth L. Chamberlain (elizabeth.chamberlain@vanderbilt.edu)

**Abstract.** Reconstructing sediment pathways in fluvial and deltaic systems beyond instrumental records is challenging due to a lack of suitable methods. Here we explore the potential of luminescence methods for such purposes, focusing on bleaching of the optically stimulated luminescence (OSL) signal of quartz sediments in a large fluviodeltaic system across time and space. We approach this by comparing residual doses of sand and silt from the modern Mississippi River channel with estimated residual doses of sand isolated from Late Holocene Mississippi Delta mouth bar and overbank deposits. Further insight is obtained from a comparison of burial ages of paired quartz sand and silt of Mississippi Delta overbank deposits. Contrasting some previous investigations, we find that the bleaching of the OSL signal is at least as likely for finer sediment as for coarser sediment of the meandering Mississippi River and its delta. We attribute this to the differences in light exposure related to transport mode (bedload vs. suspended load). In addition we find an unexpected spatiotemporal pattern in OSL bleaching of mouth bar sand deposits. We suggest this may be caused by changes in upstream pathways of the meandering channel belt(s) within the alluvial valley, or by distributary channel and coastal dynamics within the delta. Our study demonstrates that the degree of OSL signal bleaching of sand in a large delta can be highly time- and/or space-dependent. Silt is shown to be generally sufficiently bleached in both the modern Mississippi River and associated paleo-deposits regardless of age, and may provide a viable option for obtaining OSL chronologies in megadeltas. Our work contributes to initiatives to use luminescence signals to fingerprint sediment pathways within river channel networks and their deltas, and also helps inform luminescence dating approaches in fluviodeltaic environments.

## 1 Introduction

Relatively few tools presently exist to reconstruct sediment pathways within river and delta channel networks beyond instrumental (less than centennial or decadal) timescales, despite the high value of such information to human management of waterways. For example, improved knowledge of waterborne sediment pathways is a key component to delta restoration initiatives, such as river diversions aiming to mitigate land loss in deltas through delivery of sediment to the delta plain (e.g., CPRA, 2017). These engineered outlets will siphon sediment from yet-

to-be-determined depths and positions within the river channel, and so the availability and grain size of the utilized sediment will depend on its transport mode within the river (e.g., Esposito et al., 2017) as well as the location and geometry of the engineered feeder channel (e.g., Gaweesh and Meselhe, 2016). Similarly, the sustainability of such restoration strategies may hinge on the recharge timescales of river-bed sand bars, a poorly-constrained parameter.

         The luminescence signals of sediment grains provide a unique and underexplored archive for reconstructing

channel network dynamics and evolution, as they contain information regarding the transport histories of grains. . Luminescence dating (Huntley et al., 1985;Hütt et al., 1988) and its subsequent methodological advances (e.g., Cunningham and Wallinga, 2010;Galbraith et al., 1999;Murray and Wintle, 2000, 2003;Cunningham and Wallinga, 2012) have enabled new chronologies of fluvial and deltaic systems, obtained from direct measurements of the burial time of clastic sediment. Yet, relatively few studies have applied this tool to trace earth surface processes (see

Gray et al., 2017;Reimann et al., 2015;Sawakuchi et al., 2018;Liu et al., 2009;Keizars et al., 2008;Forman and Ennis, 1991). Of note, Liu et al. (2009) used feldspar thermoluminescence (TL) signals to study sand transport pathways in a fluvial and coastal system in Japan, while Reimann et al. (2015) used bleaching of multiple luminescence signals to track the dispersion of sand grains along the Dutch coast following human-emplacement for coastal engineering. Gray et al. (2017) designed a model to estimate fluvial sediment flux based on upstream-to-

downstream bleaching of sediments observed in the Loire and Mojave rivers. These studies all focused on modern deposits, rather than exploiting the archive of (pre-)historic sediments.

         Here, we explore luminescence signal resetting ("bleaching") as a means of reconstructing the fluvial pathways of sediments in Late-Holocene deposits. Bleaching of the luminescence signal by sunlight exposure of at least some of the grains within a sample upon burial is a key component of luminescence dating; in other words, the

luminescence clock of at least some grains must be zeroed shortly before or at the time of the event of interest (e.g., Wallinga, 2002). In the absence of complete resetting, sediment grains retain a residual dose acquired during

previous burial. Populations of sediment grains (e.g., sediment samples) may be well/completely-bleached and contain only zeroed grains, or may be incompletely-bleached and contain at least some grains with residual doses (Duller, 2008). Here, we further classify incompletely-bleached sediment populations as heterogeneously- (containing both zeroed grains and grains with residual doses), or poorly-bleached (containing few to no zeroed grains).

Assessing luminescence signal bleaching is important for dating applications because incomplete bleaching may lead to overestimation of the time of the most recent burial event. Incomplete bleaching is especially a concern for dating of fluvial sediment deposited within the most recent millennium, because some grains may receive little light exposure during river transport, and even small residual doses can produce highly inaccurate ages on young deposits (Wallinga, 2002). A number of studies have investigated the degree of bleaching of river deposits and sediment entrained within modern river channels, and its relationship to grain size and geography, primarily for the purpose of improving dating fidelity. This has been approached through tests of modern sediment or those of independently-constrained depositional ages. These studies have returned wide and varied results. Some have found that coarse sand is better bleached than fine sand (e.g., Olley et al., 1998, Murrumbidgee River, Australia;Truelsen and Wallinga, 2003, Rhine Meuse Delta, The Netherlands). Well-bleached silt has been identified in suspension in the Yangtze River (Sugisaki et al., 2015) and its Holocene deposits (Nian et al., 2018;Gao et al., 2018) as well as in recent (decades- to centuries-old) deposits of the Ganges-Brahmaputra Delta (Chamberlain et al., 2017). Yet, incomplete bleaching of silt has been shown for other fluvial systems such as fluvial terrace deposits of northwest China (Thompson et al., 2018) and for flood deposits of the Elbe River and a tributary, Germany (Fuchs et al., 2005).

The mechanisms that dictate the degree of bleaching of fluviodeltaic sediments are also not known to be absolute nor universal. For example, bleaching of sand may increase on average with transport distance due to sunlight exposure during temporary storage on bar surfaces (Stokes et al., 2001) or may decrease downstream due to the addition of poorly-bleached grains by tributaries or local bank erosion (McGuire and Rhodes, 2015). Sediment entrained within a river channel may be less well-bleached if temporary storage on the river banks is limited, because turbid water reduces the intensity of light exposure and restricts the light spectrum (Berger et al., 1990). Yet, in-channel transport offers an opportunity for sunlight exposure of sediment transported near the water surface, especially finer grains which are more evenly distributed in the water column (Fuller et al., 1998) or those in

turbulent systems (Gemmell, 1988). Bleaching of entrained sediments may also occur during subaerial exposure of river bar surfaces under conditions of low water discharge (Cunningham et al., 2015;Gray and Mahan, 2015). Furthermore, sediment grains in-transit may experience different bleaching than those preserved in the stratigraphic record (Jain et al., 2004) because in-transit grains have not yet reached their final destination and therefore have not necessarily undergone the full range of bleaching opportunities.

Our research aims to clarify the degree and mechanisms of luminescence signal bleaching in fluviodeltaic sediments, so that this tool may be applied to sedimentary archives to reconstruct sediment transport histories. We approach this through an investigation of residual doses of quartz sediments of the contemporary Mississippi River and associated Late Holocene deltaic deposits (Fig. 1). Using recently proposed and tested methods (Chamberlain et al., 2018b) to analyze archival data, we compare the residual quartz optically stimulated luminescence (OSL) doses of sand and silt sampled within the modern Mississippi River channel (Muñoz et al., 2018;Chamberlain et al., 2018b;Chamberlain, 2017) to those estimated from sand of multi-century to millennium-aged Mississippi Delta mouth bar (Chamberlain et al., 2018a) and overbank (Shen et al., 2015) deposits. Further insight into OSL bleaching is obtained by reanalyzing the burial ages of paired quartz sand and silt of Mississippi Delta overbank deposits (Shen et al., 2015). All combined, these data allow us to test whether OSL signal bleaching varies across time, space, grain size, and depositional environment, even within a single fluviodeltaic system.

## 2 Geologic setting

### 2.1 Mississippi River hydrology

The Mississippi River is among the largest rivers in the world in terms of catchment size, sediment, and water discharge. Its catchment includes about $3.3 \times 10^6$ km$^2$ (Milliman and Syvitski, 1992) and drains about 41% of the continental United States (Fig. 1) (Milliman and Meade, 1983). Therefore, sediment grains arriving in the Mississippi Delta may originate as far as 2,400 linear kilometers upstream and have experienced lengthy and convoluted transport (with bleaching opportunities, e.g., Stokes et al., 2001), or as near as a few meters or less from nearby river cutbanks and have experienced minimal transport (and minimal opportunities for bleaching, e.g., McGuire and Rhodes, 2015) since their last major storage event.

The hydrograph of the Mississippi River is generally highest in the spring due to snowmelt and increased precipitation in the catchment, and has multiple spring peaks with an average discharge of 25,000 $m^3$/s or more (Supplementary File, Fig. S1) (Galler and Allison, 2008). The first springtime "freshet" serves to mobilize and flush sediment from the lower reaches of the Mississippi River channel that has accumulated during preceding autumn-time low flow (less than 8500 $m^3$/s) conditions (Galler and Allison, 2008). Historical discharge records (1964-2012) for the US Army Corps of Engineers gauge at Tarbert Landing (river km 492 above Head of Passes, Fig.1) show that cumulative annual discharge is highly variable between years, and can range from around $3 \times 10^{11}$ $m^3$/yr to greater than $6 \times 10^{11}$ $m^3$/yr (Allison et al., 2014). Mud (that is, silt and clay) is the primary material transported in suspension during low flow conditions in the lower reach of the river, and is generally evenly distributed throughout the water column at all discharges (Ramirez and Allison, 2013). The mass of suspended sand in the lower reach, thought to be mobilized from lateral bars on the river bed, is minimal during low flow events and becomes similar to that of fines during the highest flow events (Allison et al., 2014). This indicates that there is a seasonal opportunity for light exposure of sands, and a year-round opportunity for light exposure of silts during transport within the river channel.

In addition to being a major river with significant variance driven by natural sources, the Mississippi is presently one of the most highly engineered river systems in the world (Kesel, 2003;Allison et al., 2012). Flow within the contemporary Lower Mississippi River is generally contained by human-made levees, which limit the degree of interaction of the channel with its floodplain and decrease the cannibalization of banks by restricting river migration (Kesel, 2003). Throughout the Holocene and prior to modification, the Mississippi River meandered freely within a series of six channel belts of unknown absolute ages (Saucier, 1994). The construction of dams as well as flood and navigation control structures in the catchment has reduced the suspended sediment load reaching the delta by reported values of 50-70 % (Blum and Roberts, 2009;Kesel, 2003), although the effects of these structures on sand transport to and within the deltaic reach has been debated (Nittrouer and Viparelli, 2014;Blum and Roberts, 2014). Similar changes in hydrology and sediment transport due to engineering have been documented in other fluviodeltaic systems (e.g., Hobo, 2015;Erkens, 2009). An investigation into bleaching that considers the residual doses of sediments of both pre-anthropogenic and present-day conditions is therefore useful because the hydrology and related luminescence bleaching opportunities of grains in the Mississippi River and other major channels worldwide may have been quite different prior to human modification of rivers.

## 2.2 Mississippi Delta stratigraphy and OSL properties

The Holocene Mississippi Delta first emerged around 7 ka, as sediment delivery to the basin outpaced regional sea-level rise (Törnqvist et al., 2004), and is composed of a series of amalgamated sediment lobes (subdeltas) fed by discrete distributary networks (Fisk, 1944). This study mainly investigates deposits of the presently 10,000 km$^2$ Lafourche subdelta (Fig. 1), active from ~1.6 to ~0.6 ka (Törnqvist et al., 1996;Shen et al., 2015;Chamberlain et al., 2018a), which has been extensively OSL dated for geologic research and therefore provides a valuable archive of data to explore bleaching of the quartz OSL signal. During its millennium of activity, the Lafourche distributary network constructed 6,000-8,000 km$^2$ of new land through progradation into a shallow bay (Chamberlain et al., 2018a), while the upstream portion of the system aggraded via deposition primarily from episodically-active crevasse channel networks (Shen et al., 2015). These crevasse channels operated as shallow off-takes, siphoning suspended material from the axial distributary channels (Esposito et al., 2017). Relatively coarse and homogenous mouth bar sand deposits characterize the prograded portion of the subdelta (Chamberlain et al., 2018a). A patchwork of generally finer-grain crevasse splay and natural levee deposits, which may overlie progradational facies or peat, characterize the near-channel overbank depositional environment (Törnqvist et al., 2008;Shen et al., 2015;Esposito et al., 2017;Mehta and Chamberlain, 2018). Discharge was shared between the Lafourche and Modern (Balize) subdeltas beginning with Modern subdelta initiation at ~1.4 - 1.0 ka and continuing until Lafourche subdelta abandonment at ~0.6 ka (Hijma et al., 2017), although the exact timing and nature of the discharge split is unknown.

Previous research applying OSL dating to Lafourche subdelta deposits mainly relied on the measurement of small-diameter aliquots (that is, numerous subsamples for each sample, each containing ~23-108 grains) of quartz sand in combination with the application of minimum age models (Galbraith et al., 1999;Cunningham and Wallinga, 2012) to extract paleodoses, because equivalent dose ($D_e$) distributions suggested that at least some of the fluvial deposits in this setting were not completely bleached (Shen et al., 2015;Chamberlain et al., 2018a). These approaches were found to yield internally consistent OSL ages which agreed with radiocarbon constraints obtained from prior dating of underlying peat (Törnqvist et al., 1996), whereas the application of the Central Age Model (CAM) (Galbraith et al., 1999) was found to overestimate the OSL age of some samples (Chamberlain et al., 2018b). Shen and Mauz (2012) found that the subtraction of an early background interval (Cunningham and Wallinga, 2010) produced more accurate and younger OSL ages for contemporary deposits associated with the nascent Wax Lake

Delta of the Mississippi Delta (Fig. 1), also suggesting incomplete OSL bleaching of Mississippi Delta grains. Results of their study were validated with independent chronology from historical records. Shen et al. (2015)

employed late background subtraction for dating overbank sands and silts, and showed that late-background-subtracted luminescence ages for paired silt and sand fractions extracted from the same overbank samples agreed within 2σ, indicating that silt may be a viable option for OSL dating of Late Holocene Mississippi Delta deposits (e.g., Muñoz et al., 2018).

**3 Methods**

**3.1 Compilation of archival data**

This study uses quartz OSL data compiled from previous investigations of contemporary sediments of the Mississippi River (Muñoz et al., 2018;Chamberlain et al., 2018b;Chamberlain, 2017) and associated prehistoric

deltaic deposits (Chamberlain et al., 2018a;Shen et al., 2015)  (Fig. 1, Table S1).

Modern Mississippi River bedload (Chamberlain et al., 2018b) and suspended load (Chamberlain, 2017;Muñoz et al., 2018) sediments were sampled at Bonnet Carre Upstream 2 (BCU2), a site 221 river kilometers above the Mississippi River mouth at Head of Passes (Fig. 1). This site corresponds to the AboveBC2 site in Allison et al. (2013). Sampling took place in the Mississippi River channel center during high-flow conditions of 18,320

$m^3$/s in May, 2014, when the channel depth at BCU2 was 21.9 m. Suspended sediment samples (n=5) were captured in 5 L Niskin bottles at 0% (0 m), 25% (5.5 m), 50% (11.0 m), 75% (16.4 m), and 90% (19.7 m) water depths and a bedload sediment sample (n=1) was captured with a grab sampler. All samples were covered during and following retrieval to prevent light-exposure. OSL results obtained from the suspended samples were previously fully documented in a doctoral dissertation (Chamberlain, 2017), and data from the fine (4-20 μm) silt fraction were

published by Muñoz et al. (2018) without details regarding the sampling location and approach, and analytical aspects. Here, we present the essential details of these five samples, including OSL data for coarse (45-75 μm) suspended silt (n=2).

To investigate bleaching of older sediments, we revisited samples of Late Holocene Mississippi Delta sediments ranging in age from 1.6 - 0.6 ka, previously collected and measured by Shen et al. (2015) (overbank

deposits, n=23) and Chamberlain et al. (2018a) (mouth bar deposits, n=17). The overbank samples were collected in

the upper reaches of the Lafourche subdelta, from deposits that formed through aggradation primarily fed by crevasse splay networks. The mouth bar samples were collected in the lower reaches of the Lafourche subdelta, and capture the growth of this portion of the system as it expanded radially into open water. Details of these paleo-deposit samples and of the growth history of the Lafourche subdelta are available in their primary publications.


**3.2 OSL measurements, age modeling, and residual dose calculations**

Samples were prepared following standard procedures which are described in the primary publications and were generally consistent across datasets. Measurements of small-diameter (~1.2 mm) sand aliquots (containing ~23-108 grains per aliquot, see Chamberlain et al., 2018b) and ~2 mg/disk silt aliquots (containing over 1 million grains per

aliquot) were conducted using standard single-aliquot regenerative dose protocols (Murray and Wintle, 2000, 2003), also described in the primary publications. Relatively low preheat temperatures (200 - 220 $^{o}$C) were adopted to avoid thermal transfer. To ensure consistency across the archival luminescence data repurposed here, the original output luminescence data (BIN/BINX-files, generated through luminescence measurements using Risø readers) were reanalyzed using standardized approaches (Chamberlain et al., 2018b), which most-importantly included the

subtraction of an early background interval from the integrated initial OSL signal to enhance the relative contribution of the most readily-bleached quartz fast component to the net OSL signal (Cunningham and Wallinga, 2010). For $D_e$ estimation of individual aliquots, uncertainties related to instrumental reproducibility (1.5 %) and growth curve fitting were included. Hence overdispersion estimates obtained on these datasets using the CAM model reflect all non-explained scatter (due to e.g. between-grain dose rate heterogeneity, heterogeneous bleaching,

but also that related to inaccuracies in $D_e$ estimation due to uncorrected sensitivity changes).

Estimating residual doses of in-transit modern sediments is fairly straightforward, because these should yield a zero $D_e$ when completely bleached. The residual doses of modern river sand samples were therefore estimated as the doses obtained from the CAM. For completeness, doses of the modern river sand samples were also modeled with the unlogged version (Arnold et al., 2009) of the bootstrapped (Cunningham and Wallinga, 2012)

Minimum Age Model (Galbraith et al., 1999) (bootMAMul) and estimated with a mean and standard error on the mean (henceforth referred to as standard error), but these results were not used for analysis. The residual doses of modern river silt samples were estimated with a mean and standard error. We applied this approach because we have previously observed that the CAM preferentially weights the higher dose aliquots (due to lower relative uncertainty)

and thus overestimates the central dose of very young and well-bleached sediment (Chamberlain et al., 2017). Also for completeness, we modeled the central doses of these silts with the CAM, but did not use these results for analysis.

In the absence of independent age control for individual samples (e.g., historical records of deposition), the estimation of the residual dose of paleo-deposits is not straightforward. Here, we estimated the residual doses of Late Holocene deposits as the difference in $D_e$s obtained with the CAM ($D_{e,CAM}$) and the bootMAM ($D_{e,bootMAM}$) (Chamberlain et al., 2018b) (Fig. 2). This novel approach assumes that 1) the CAM provides an accurate value for the central dose, and 2) the bootMAM captures the true paleodose of the sample, representative of its depositional age.

Age modeling for the paleo-deposit sand samples was revisited by Chamberlain et al. (2018b) using the bootstrapped (Cunningham and Wallinga, 2012) Minimum Age Model (Galbraith et al., 1999) (bootMAM). This study also employed a novel approach to assign the input and uncertainty on the overdispersion ($\sigma_b$) parameter using the existing dataset itself (Chamberlain et al., 2018b), through the following steps: 1) samples were grouped by grain size, 2) $D_e$ datasets of samples for the grain size group with the greatest number of samples were input to the CAM, yielding the overdispersion and uncertainty of each sample, 3) these overdispersion data were input to the bootMAM with the $\sigma_b$ parameter set to [0 0], outputting the $\sigma_b$ value representative of the overdispersion of the best-bleached samples for the selected grain size group, and 4) $\sigma_b$ for the other grain size groups was obtained from this output value by correcting for the number of grains per disk following Cunningham et al. (2011). Ages obtained through this approach for mouth bar and overbank deposits of Bayou Lafourche provided highly consistent datasets (Chamberlain et al., 2018b), in agreement with independent age constraints (Törnqvist et al., 1996). Hence, depositional ages reconstructed with the bootMAM are expected to be valid, while doses obtained with the CAM are similar to the mean and standard error (Supplementary File, Table S1, Fig. S2). Thereby both assumptions outlined above are satisfied, and the remnant dose obtained from difference in $D_e$s obtained with the CAM ($D_{e,CAM}$) and the bootMAM ($D_{e,bootMAM}$) provides a robust measure of the degree of bleaching.

The doses of silt isolated from paleo-deposits were determined as a mean and standard error, and we also report central doses obtained with the CAM (Table S2), which were not used for analysis. The mean has been shown to yield identical doses as the CAM for silt deposits greater than a few hundred years in depositional age (Chamberlain et al., 2017), consistent with our findings here (Table S2). To test bleaching of Late Holocene silt, we

compare the silt ages obtained from the mean to bootMAM ages of sand isolated from the same sample, because 1) within-aliquot averaging disqualifies the use of a minimum age model, thus internal comparison of $D_{e,CAM}$ - $D_{e,bootMAM}$ is not possible for the silt fraction, 2) we found the bootMAM sand ages to provide a robust estimate of burial age, and 3) remnant age is preferred over remnant dose for silt, as silt and sand grains within the same sediment matrix experience different dose rates, rendering a comparison of doses less useful.

### 3.3 Dose rate and residual age estimation

Remnant doses preserved in grains upon burial have little direct relationship with the dose rate of the matrix from which the grains are ultimately isolated for luminescence dating, although the dose rate of this matrix is often used to determine the residual age. The bulk sediment characteristics and geological context (e.g., radionuclide activity concentrations, cosmogenic exposure, water content) under which the residual doses were acquired are generally unknown. For this reason, we prefer to use residual dose rather than residual age to describe the bleaching of sediments if possible. Hence we use residual dose for all sand extracts (modern and paleo-deposit), and for modern river silt extracts (see section 3.2). Approximations of residual age are also discussed for the sand extracts and modern river silt extracts, as it relates to potential inaccuracies in burial age estimates if incomplete bleaching is not adequately addressed. These are informed by average (± standard deviation) dose rates of 2.43 ± 0.35 Gy/ka for sand, and 2.96 ± 0.25 Gy/ka for silt sampled within the Lafourche subdelta (Shen et al., 2015;Chamberlain et al., 2018a).

Ages calculated for the comparison of sand and silt fractions isolated from the same sample used dose rates particular to those samples, presented in Shen et al. (2015). These ages were calculated by dividing the paleodose of each sample by its dose rate, taking both random and systematic uncertainties into account, and propagating uncertainties in quadrature. All dose rates from Shen et al. (2015) were updated here to use the radionuclide conversion factors of Guérin et al. (2011) (Supplementary File, Table S.1). Other dose rate details can be found in the original publications.

### 4 Results and interpretation

### 4.1 Residual doses of modern river sediments

Residual doses of all samples are provided in the Supplementary File, Table S1. Modern river sediments show a trend of increasing residual dose with both grain size and sampling depth below the water surface (Fig. 3). We found that residual doses of modern river silt, moving in suspension within the channel, are very low regardless of water depth (note the logarithmic scale of Fig. 3). These ranged from $0.027 \pm 0.001$ to $0.135 \pm 0.013$ Gy for the 4-20 µm grains, with a mean value of $0.078 \pm 0.044$ Gy, similar to the "bulk" $D_e$ values reported by Muñoz et al. (2018). Our

reported mean residual dose, obtained through subtraction of an early background interval plus other methods described in Chamberlain et al. (2018b), corresponds to an estimated residual age of 12-41 years (1σ range).

Of the two coarse silt (45-75 µm) suspended load samples, only the deeper (19.7 m) sample (BCU2 I-5) produced a measurable quartz luminescence signal, while the coarse silt fraction of the shallower sample (BCU2 I-3, at 11.0 m depth) was not sufficiently luminescent and will not be discussed further. The residual dose of BCU2 I-5

was $0.227 \pm 0.149$ Gy (26-127 years), suggesting that bleaching of coarser silt transported deeper in the water column may be less complete than bleaching of finer silt moving in more shallow suspension (Fig. 3). However, this tentative suggestion will need further confirmation, as the comparison is based on a single sample and the results on the fine and coarse silt fraction for this sample agree within errors.

Residual doses of modern river fine silt appeared to be slightly greater with depth in the channel (Fig. 3).

This may suggest some stratification of the water column. Alternatively, and more likely, the apparent trend may reflect different grain-size distributions within the analyzed fractions. Although the same fraction (4-20 µm) was prepared for each of the suspended samples, the deeper samples are more likely than the shallow samples to contain relatively large silt grains within this range, because coarser grains are more likely to be transported near the river bed than finer grains. The lower samples are therefore more likely to contain material that experienced less light

exposure. This interpretation is consistent with the observation that the 45-75 µm suspended sample (BCU2 I-5) appeared less completely bleached than all the finer, 4-20 µm suspended silt samples (BCU2 I-1,2,3,4,5) (Supplementary file, Table S1).

By contrast, both grain size fractions of modern river bedload sand (BCU2 I-6) appeared to be heterogeneously-bleached. The residual dose of the 125-180 µm fraction of BCU2 I-6 was $1.62 \pm 0.29$ Gy. This

corresponds to a 0.51-0.82 ka estimated residual age. A bootMAMul $D_e$ of $0.027 \pm 0.051$ Gy indicated that this grain size fraction contained some well-bleached quartz grains capable of producing an accurate luminescence age. The residual dose of the 180-250 µm fraction of BCU2 I-6 was $10.5 \pm 1.7$ Gy, corresponding to a 3.40-5.25 ka estimated

residual age. A bootMAMul $D_e$ of 0.79 ± 0.53 Gy indicated that this grain size fraction contained very few, if any, well-bleached quartz grains. Nevertheless, the estimate agrees with the expected zero dose at the 2σ level, demonstrating the ability of the bootMAM model to provide accurate (yet imprecise) results for highly heterogeneously-bleached samples. We note that some aliquots of the coarse sand fraction provided $D_e$s of more than 25 Gy (> 10 ka residual age), indicating that some coarser sand grains transported by the modern Mississippi River originated from pre-Holocene deposits with limited light exposure during transport.

**4.2 Residual doses of late Holocene deposits**

As there is uncertainty on the $D_{e,CAM}$ and $D_{e,MAM}$ values from which residual doses were calculated, there is also uncertainty on the residual doses. The bleaching of each Late Holocene-aged sample was classified by its minimum residual dose, defined as the residual dose minus 1σ uncertainty. This means that some samples classified as well bleached may have possessed small residual doses. Sand isolated from mouth bar and overbank deposits ranged from well- to heterogeneously-bleached for both depositional environments (Fig. 4). This was indicated by residual doses, calculated as $D_{e,CAM}$ - $D_{e,bootMAM}$, ranging from zero to 2 Gy (and a single estimate of greater than 3 Gy). These values correspond to residual ages estimated to be in the range of 0 - 1 ka. Mouth bar deposits had a smaller proportion of well-bleached sand samples (29%), while overbank deposits contained a greater proportion of well-bleached sand samples (48%). Bleaching was more complete for samples with paleodoses less than about 2.3 Gy (Fig. 4). Above 2.7 Gy, mouth bar sand was found to be heterogeneously-bleached with considerable (>0.5 Gy) residual doses, while overbank sand of similar $D_e$s ranged from well to heterogeneously bleached (Fig. 4).

**4.3 Bleaching by grain size**

Among all samples, we observed a trend of increasing residual dose with increasing median grain size (Fig. 5), suggesting that coarser sand may be the least likely grain size to be completely bleached in this system. Still, each sand grain size fraction also contains some well-bleached samples, indicating that sand grains of all investigated sizes could be bleached prior to preservation. As discussed above, the 180-250 μm fraction of the river bedload sample (BCU2 I-6) yielded an exceptionally high residual dose of more than 10 Gy. While results for this grain size fraction fit the observed trend of bleaching degree with grain size, they are informed by only one sample of sediment that was still in-transit in the river channel when captured and may not be representative of bleaching of these

coarser grains, both moving in the channel and preserved in the stratigraphic record (e.g., Jain et al., 2004). For these reasons, we caution against over-interpreting the results of BCU2 I-6, and the coarser fraction of the modern river bedload sample is omitted from Fig. 5.

**4.4 Temporal trends in bleaching**

Surprisingly, the bleaching of mouth bar sand showed a strong temporal trend (Fig. 6A). All mouth bar sand samples (n=7) older than ~ 1.2 - 1.1 ka possessed relatively high residual doses, ranging from about 1 Gy to more than 3 Gy. Bleaching of sand isolated from mouth bar deposits younger than 1.1 ka (n=10) was much improved, with all samples yielding residual doses less than 0.1 Gy within uncertainty. Overbank sand also showed a trend of better bleaching with time, although there remained significant variability in the degree of bleaching, with both well- and heterogeneously-bleached overbank sands of all ages (Fig. 6A).

**4.5 Spatial trends in bleaching**

As the Lafourche subdelta expanded radially (Chamberlain et al., 2018a), the temporal trend in bleaching of mouth bar deposits is also reflected spatially (Fig. 6B). Mouth bar deposits in the upstream reaches (above ~105 river km) are less well bleached, compared to those in downstream parts. To determine whether bleaching may occur during the overbank/crevasse process, residual doses of overbank sand at the PV, EF, and NV sites were plotted against distance to the present-day bank of Bayou Lafourche. This test revealed no spatial trends within the overbank sands (Supplementary File, Fig. S3), although we do note that overbank sand samples from different depths within the same borehole tend to have similar degrees of bleaching (Supplementary File, Table S1 and Fig. S4).

**4.6 Bleaching of Late Holocene silt inferred from sand/silt pairs**

Good agreement was found between the majority (n=5) of sand and silt pairs dated from the same overbank samples (n=7) (Fig. 7), broadly consistent with the findings reported by Shen et al. (2015). Silt ages (obtained using the mean) scattered both higher and lower than sand ages (obtained using the bootMAM), indicating that these silts were generally sufficiently bleached for dating (Fig. 7A). Two samples, PV I-4 and PV I-5, produced silt ages that exceed sand ages by ~ 400 and 600 years respectively. The age overestimation by silt may be due to poor bleaching of silt-sized quartz (Shen et al., 2015). Alternatively, we identified evidence that OSL signals of these two samples

contained a contribution from feldspar, which is less readily bleached than the fast-component signal of quartz

(Godfrey-Smith et al., 1988;Wallinga, 2002) (Supplementary File, Fig. S5).

By coincidence, the samples selected by Shen et al. (2015) for the paired sand/silt analysis featured sand

that we mainly classified as well-bleached, with little difference between sand ages obtained with the CAM and the

bootMAM  (Supplementary File, Fig. S6). This manifests in similar ages obtained with the mean dose of the silt

fraction and the CAM dose of the sand fraction (Fig. 7B). It is possible that greater differences between bleaching of

sand and silt could be identified if this test was performed on sediment pairs extracted from deposits with

heterogeneously-bleached sand.

## 5 Discussion

### 5.1 Interpretation of bleaching trends and implications to luminescence signal bleaching

It is little wonder that universal trends in bleaching of fluviodeltaic sediments have not yet been identified,

considering the complex and numerous pathways river sediments may take prior to deltaic deposition and the natural

variability among river systems in general. This study, which focused on one large meandering river and its deltaic

deposits across time, identified lower average quartz OSL residual doses for finer sand grains than for coarser sand

grains. This trend was observed both for in-transit sediment within the river and for sediments preserved within

deltaic deposits (Figs. 3 & 5). Our findings with regard to grain-size-dependent sand bleaching are different from

those of studies conducted in other systems (Truelsen and Wallinga, 2003;Olley et al., 1998), which featured smaller

primary channels and included more samples of coarser grain sizes than investigated here.

We found that fine silt, moving in suspension within the modern river channel, was more completely

bleached than sand moving as bedload (Fig. 3), and that bleaching of silt was also generally sufficient in river

sediments deposited prior to human engineering of the system (Fig. 7). This is consistent with recent studies of other

contemporary large river systems within their deltaic reaches (Sugisaki et al., 2015;Chamberlain et al., 2017), yet

different again from studies of smaller and/or source-proximal rivers and their deposits (Fuchs et al.,

2005;Thompson et al., 2018). As many large rivers are well known to be turbid (e.g., the "Muddy" Mississippi,

Morris, 2012;Gramling, 2012;Rutkoff and Scott, 2005), it is possible that turbulence within large and lengthy

channels offers sufficient opportunities for bleaching of the finest material moving in suspension, via the constant

upwelling of the sediment-laden river water.

Bleaching of mouth bar sand (75-125 and 125-180 μm), which generally includes the coarsest material transported by a distributary system (Wright, 1977), increased in time (Fig. 6A) and coastward (Figs. 6B & 8). Temporal and spatial trends coincide, due to radial growth of the delta through bayhead delta progradation (Chamberlain et al., 2018a). The links between both make it difficult to parse the relationship of bleaching to distance versus time.

We offer a few plausible explanations for the spatiotemporal trend in bleaching of mouth bar sand within the Lafourche subdelta (Figs. 6 & 8):

1)  The primary alluvial channel is known to have avulsed a number of times throughout the Late Holocene (Saucier, 1994;Chamberlain et al., 2018a) and to have migrated via meandering within its channel belts, thereby occupying different pathways within the Lower Mississippi Valley (well upstream of the delta). The timing of channel belt avulsions and meander pathways is not well known.  It is possible that a relatively landward avulsion (450-700 linear km inland, see Chamberlain et al., 2018a) or divergence in a meandering pathway of the river within one channel belt circa 1.2 - 1.1 ka may have positioned the river in such a way that it mobilized younger deposits, for example by reworking late Holocene channel-belt deposits rather than eroding Pleistocene terrace deposits. Recently-bleached sediments would require less light exposure during transit in the river system to become well-bleached upon arrival and deposition in the delta.

2)  Alternatively, the abrupt change in bleaching of mouth bar sand may be linked to hydrologic changes within the delta itself, associated with the activation of the Modern (Balize) subdelta circa 1.4 - 1.0 ka (Hijma et al., 2017). For example, after 1.2 - 1.1 ka much of the bedload may have been rerouted toward the Modern (Balize) subdelta, causing suspended-load transport during high-flow events to be the more dominant mode of sand-delivery to the lower reaches of the Lafourche subdelta. Although under this scenario, the overbank deposits could be expected to be better bleached, because these are sourced to suspended material (Esposito et al., 2017).

3)  Additionally, decreased discharge in Lafourche distributaries (due to a partial avulsion to the modern route) or enhanced exposure as Lafourche channel tips prograded seaward and outside of the shelter of the pre-Lafourche bay may have allowed marine processes to gain importance, potentially altering turbulence, turbidity, salinity, and/or suspension times of sediment at the mouths of Lafourche distributaries. Yet, the

residual doses for overbank deposits also show a change in degree of bleaching around 1.1 ka. Although this trend is less clear compared to the mouth bar deposits, it suggests that sediment reworking at the river mouth is not the only explanation.

It is also plausible that these drivers operated in tandem; an avulsion of the alluvial channel may have driven delta-lobe switching circa 1.2 - 1.1 ka, and subsequent mobilization of younger sand upstream plus hydrologic changes at

the Lafourche channel mouths that supported more complete OSL signal bleaching. There are not sufficient data at present to test these hypotheses. Bleaching of mouth bar sand was not found to correlate to depth within the deposit (Fig. 9A), suggesting that improved bleaching was not related to reworking of mouth bar surfaces nor bioturbation, which could be expected to produce greater bleaching for shallower deposits.

Bleaching of overbank deposits was also not found to be improved at shallower depths (Fig. 9B), or with

proximity to the trunk channel (Supplementary File, Fig. S3 & S4). Other possible trends in bleaching of overbank sand merit further testing. The degree of bleaching of overbank sand may be linked to opportunities for bleaching during or immediately after deposition (e.g., Cunningham et al., 2011), or even to the time of year (and therefore water velocity and turbulence within the primary channel, e.g., Allison et al., 2014) that deposits formed.

The discussions above highlight the limitations of our study. Although this is the largest dataset used to

examine bleaching of fluviodeltaic sediment to our knowledge, there remain a number of questions that are unanswerable with the present data. For example, we observed highly-heterogeneous bleaching of the coarsest grain size of modern river bedload sand. Yet, it is unclear whether this would be the case for similar fractions sampled from different locations in the river or at different times of the year. In addition to small sample numbers for some groups, it is also difficult to parse some specific processes that drive bleaching due to confounding variables. This is

demonstrated by our discussion of the spatiotemporal trend in bleaching of mouth bar sands.

### 5.2 Implications to luminescence dating

Regardless of the limitations, our study is unique in the number and diversity of samples used to test bleaching, and it therefore makes strides toward capturing the variability of OSL signal bleaching of sediments in

the Mississippi River and Delta, and thus the natural (and anthropogenic) complexity of fluviodeltaic systems. Our results clearly show that bleaching of fluviodeltaic sediment can vary greatly by time, space, grain size, and/or depositional environment, even within a single river-delta system. Had our study only investigated a small subset of

the data herein, we could have easily arrived at different conclusions with regard to bleaching by grain size. For example, a comparison of 75-125 μm overbank sands to only the 125-180 μm mouth bar sands younger than 1.1 ka would have indicated that coarse grains are the best bleached sand fraction in the Mississippi Delta, while a comparison to only the mouth bar sands older than 1.1 ka would have yielded the opposite finding. The complexity of our dataset demonstrates that caution is needed when modern analogues are used to infer the degree of bleaching of paleo-deposits.

**5.3 Sediment fingerprinting and relevance to delta restoration**

Despite the inherent complexity of river networks, the science of luminescence dating is advancing through the use of luminescence signals to fingerprint fluvial sediments and reconstruct the routing of grains (e.g., Sawakuchi et al., 2018;McGuire and Rhodes, 2015). Our study demonstrates how luminescence signal bleaching may link to transport histories and/or the fluvial conditions under which grains are deposited, and gives insight into the last light exposure of sediment grains within a river channel. Such information is of high relevance to sustaining the Mississippi Delta and perhaps other deltas by engineered river diversions (e.g., CPRA, 2017), because the success of diversions will rely in part on their feeder channel's ability to mine sediment from suspended and/or bedload material within the river. For example, it has been proposed that locating diversions near sand bars on the river bed may maximize sand capture, thereby supplying the coarsest material needed to build a solid substrate of new land (e.g., Allison and Meselhe, 2010;Nittrouer et al., 2012;Meselhe et al., 2012). The residence times of river-bed bars and their ability to recharge are not well known, yet could be probed through estimates of the OSL residual doses of the bar sands. The methodology applied herein may thus provide a foundation for future work relevant to delta restoration.

**6 Conclusions**

This study presents the first application of a recently proposed and tested method to quantify bleaching of the OSL signal of sand grains, making use of the difference in doses obtained through central and minimum age models. We also test the bleaching of fine-grained sediments, through measurement of modern analogues and through sand-silt pairs isolated from the same deposits. Through our analysis of a large and diverse dataset of Mississippi River

bedload and suspended load sediments, and sediments of Late Holocene Mississippi Delta deposits, we arrive at the following conclusions:

- OSL signal bleaching of sand within a large delta can be highly temporally and/or spatially variable. Inferences about the degree and mechanisms of bleaching of fluviodeltaic sediments should therefore be drawn from large datasets. For dating purposes (e.g., establishing overdispersion of well-bleached samples
for age model input), it is best if such datasets include samples from the time interval, depositional environment, and region of interest.

- Quartz silt extracted from Late Holocene Mississippi Delta deposits and from suspension within the modern Mississippi River were generally well-bleached, consistent with previous findings in other large fluviodeltaic systems. The upwelling of turbid water may therefore play a significant role in bleaching of
suspended sediment in large rivers, and quartz silt should be further tested as a viable option for luminescence dating in megadeltas.

- Although there are many unknowns with regard to processes that drive the luminescence signal bleaching of river sediment, our research demonstrates the potential of this rapidly-advancing tool to yield insight into the routing of sediments through fluvial systems, which is of relevance to delta restoration initiatives.


**7 Code Availability**

     Bootstrap scripts for age modeling are available through the Netherlands Centre for Luminescence dating website (https://www.ncl-geochron.nl/en/ncl-geochron/Service.htm).

**8 Data availability**

     This research is based on archival data; references are made to original publications.

**9 Author contributions**

     The research was designed by EC and JW. Analyses were conducted by EC, with input from JW. Both
authors contributed to the interpretation of results and manuscript synthesis.

**10 Competing interests**

The authors declare that they have no conflict of interest.

**11 Acknowledgements**

We thank Mead Allison and Michael Ramirez for providing information regarding river conditions during

sampling of the modern river sediments, and for enabling their collection. We thank Susan Packman, Mhairi

Birchall, Zhixiong Shen, and Barbara Mauz for laboratory support. This manuscript benefitted from reviews by

Alastair Cunningham and Sebastian Kreutzer. The work was improved by comments on an earlier draft by Torbjörn

Törnqvist, Steve Goodbred, Mead Allison, Barbara Mauz, and Zhixiong Shen, and by discussions with Tony

Reimann and participants at the 2017 LED conference.

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

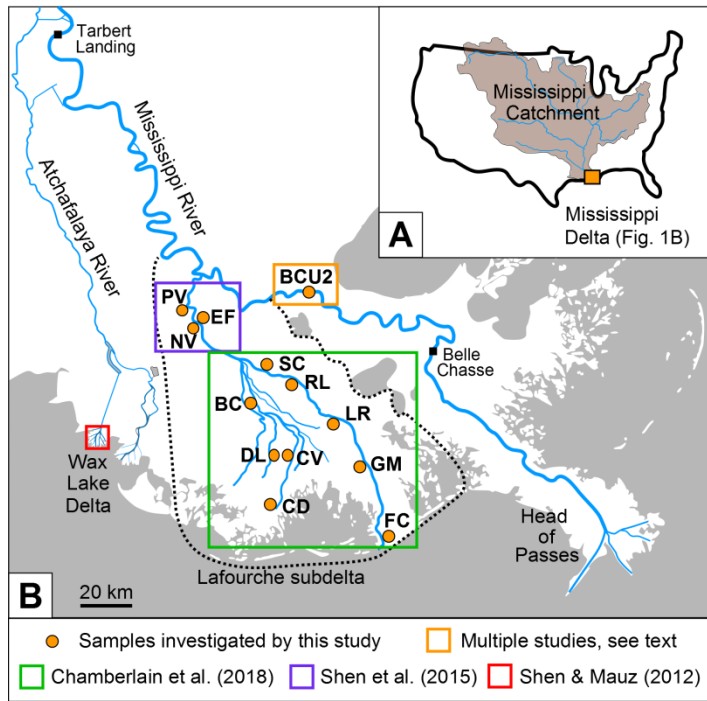

**Figure 1. The Mississippi Delta and catchment (A), and locations of contemporary Mississippi River and Lafourche subdelta samples used for this study and their primary references, plus the locations of previous research in the Wax**

**Lake Delta and of river gauge stations (B).  See Sect. 3.1 Compilation of Archival Data for the primary references of the modern river samples (orange box). Modified from Chamberlain et al. (in press).**

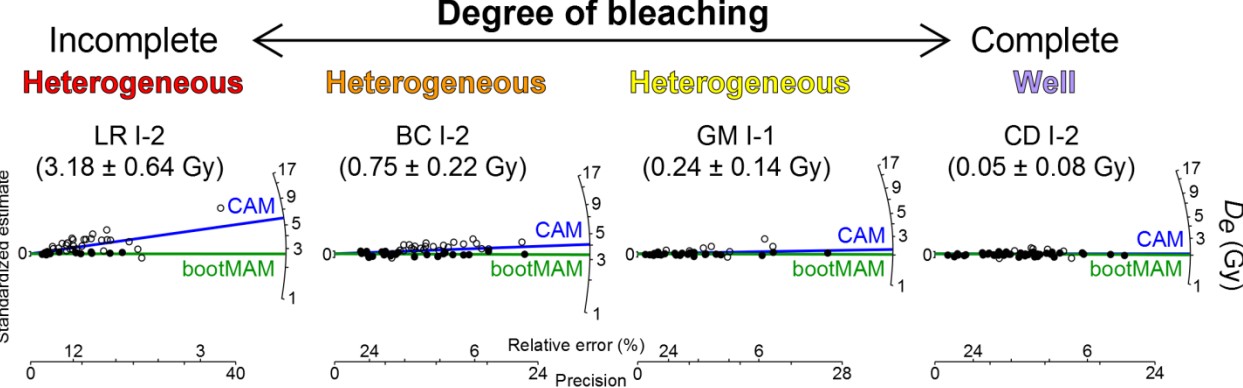

**Figure 2. Radial plots of four mouth bar sand samples provide an example of our approach to assessing the degree of bleaching of late Holocene deposits. Our assessment is based on the residual dose, given in parentheses, obtained from the difference in equivalent doses ($D_e$s) estimated with the bootstrapped Minimum Age Model (bootMAM) and Central Age Model (CAM). Filled data points represent aliquots for which the $D_e$ estimate agrees with the sample $De$ obtained from bootMAM within 2σ uncertainty. Adapted from Chamberlain et al. (in press).**


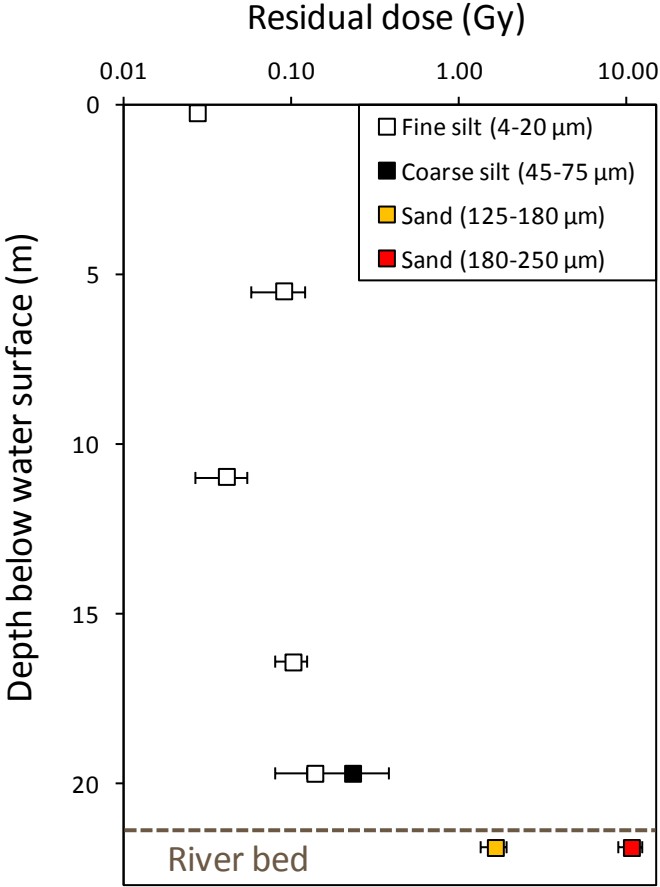

**Figure 3. Residual doses of quartz silt and sand from sediments in transit in the modern Mississippi River, with sample depth in the river channel.**

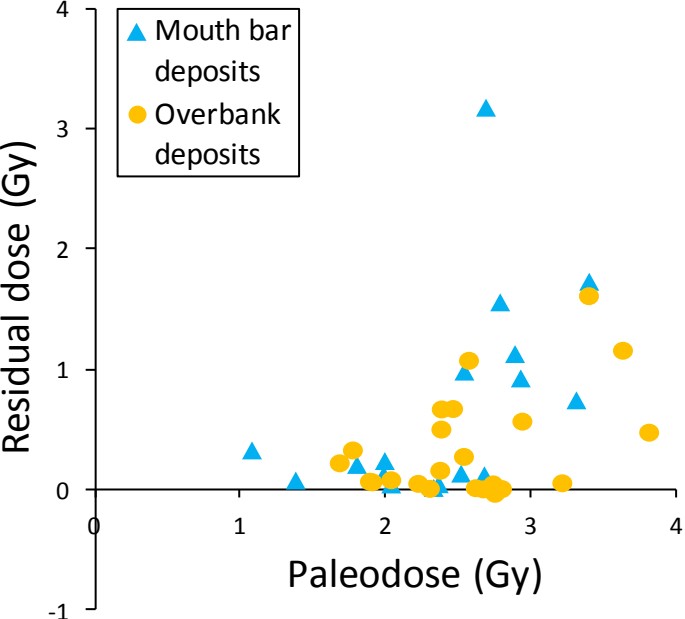


**Figure 4. Residual doses calculated as $D_{e,CAM} - D_{e,bootMAM}$ versus the paleodose estimated as $D_{e,bootMAM}$ for mouth bar and overbank deposits of the Lafourche subdelta. Uncertainties, not shown here due to the high density of data points, are given in Table S1.**

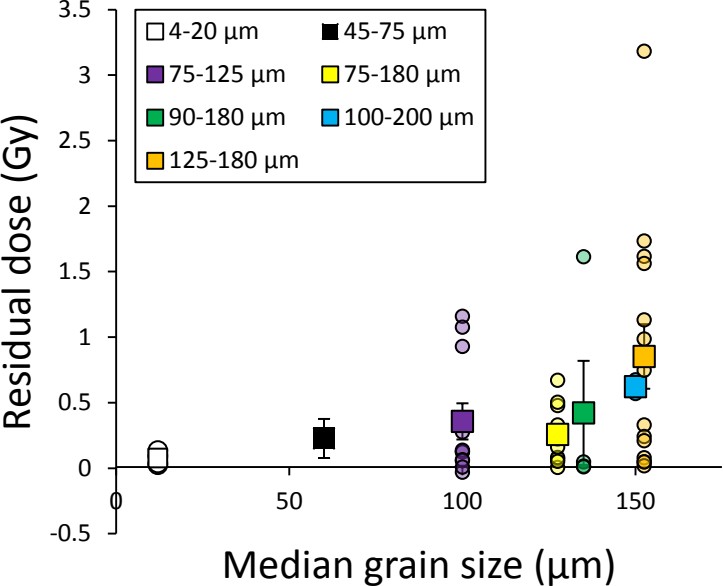


**Figure 5. Mean (boxes) and individual (circles) residual doses by median grain size (see legend) for silt and sand samples of all depositional environments. Data are not shown for the 180-250 μm fraction, which consisted of only one sample and would plot outside the graph.**

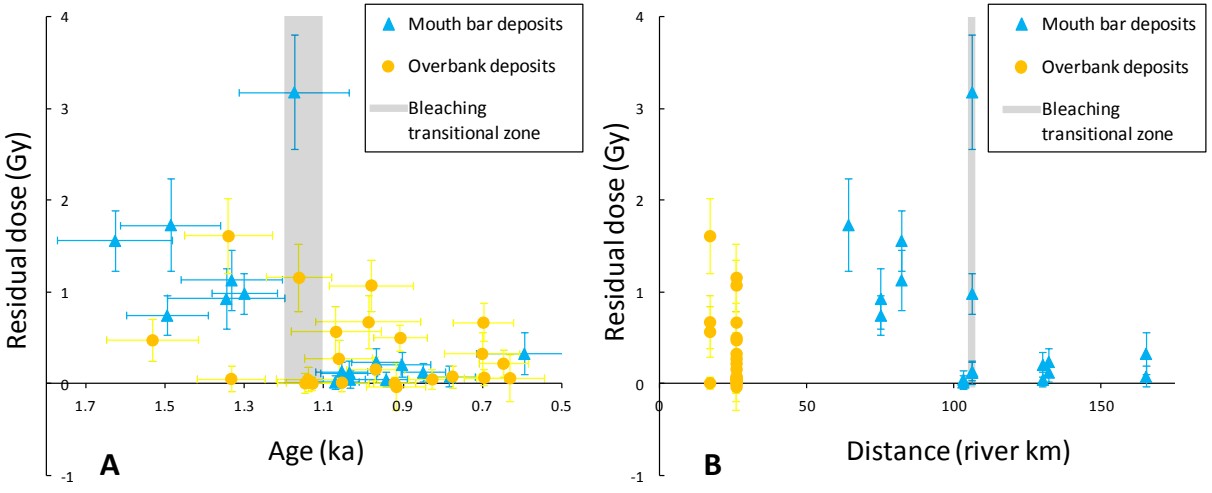

**Figure 6. Bleaching of sands isolated from mouth bar and overbank deposits with (A) burial age obtained using the bootMAM OSL approach, and (B) distance seaward, in river kilometers relative to the junction of the modern river channel and Bayou Lafourche. The shaded region indicates the transition zone from heterogeneously- to well-bleached mouth bar deposits circa 1.2-1.1 ka, or around 100 river km seaward of the junction.**

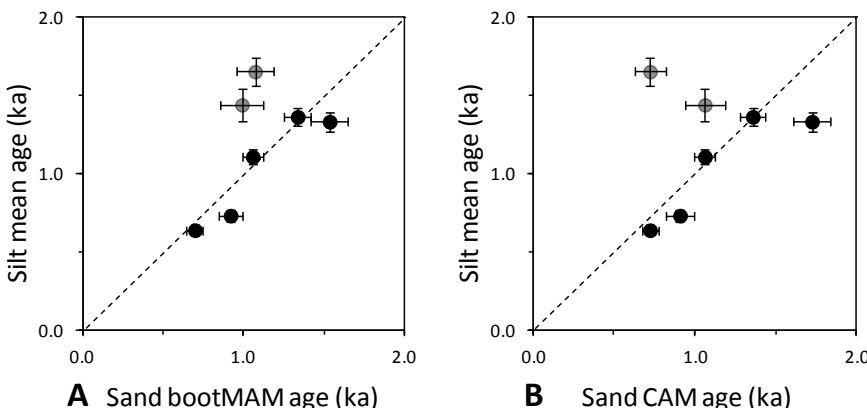

**Figure 7. Comparison of mean ages of silt fractions with ages obtained using the (A) bootMAM and (B) CAM on paired sand fractions, isolated from the same samples (n=7) of overbank deposits. Samples were collected by Shen et al. (2015) and reanalyzed here using early background subtraction plus other criteria. Gray circles indicate PV I-4 and PV I-5, two samples possibly affected by feldspar contamination of the silt fraction or containing poorly bleached silt.**

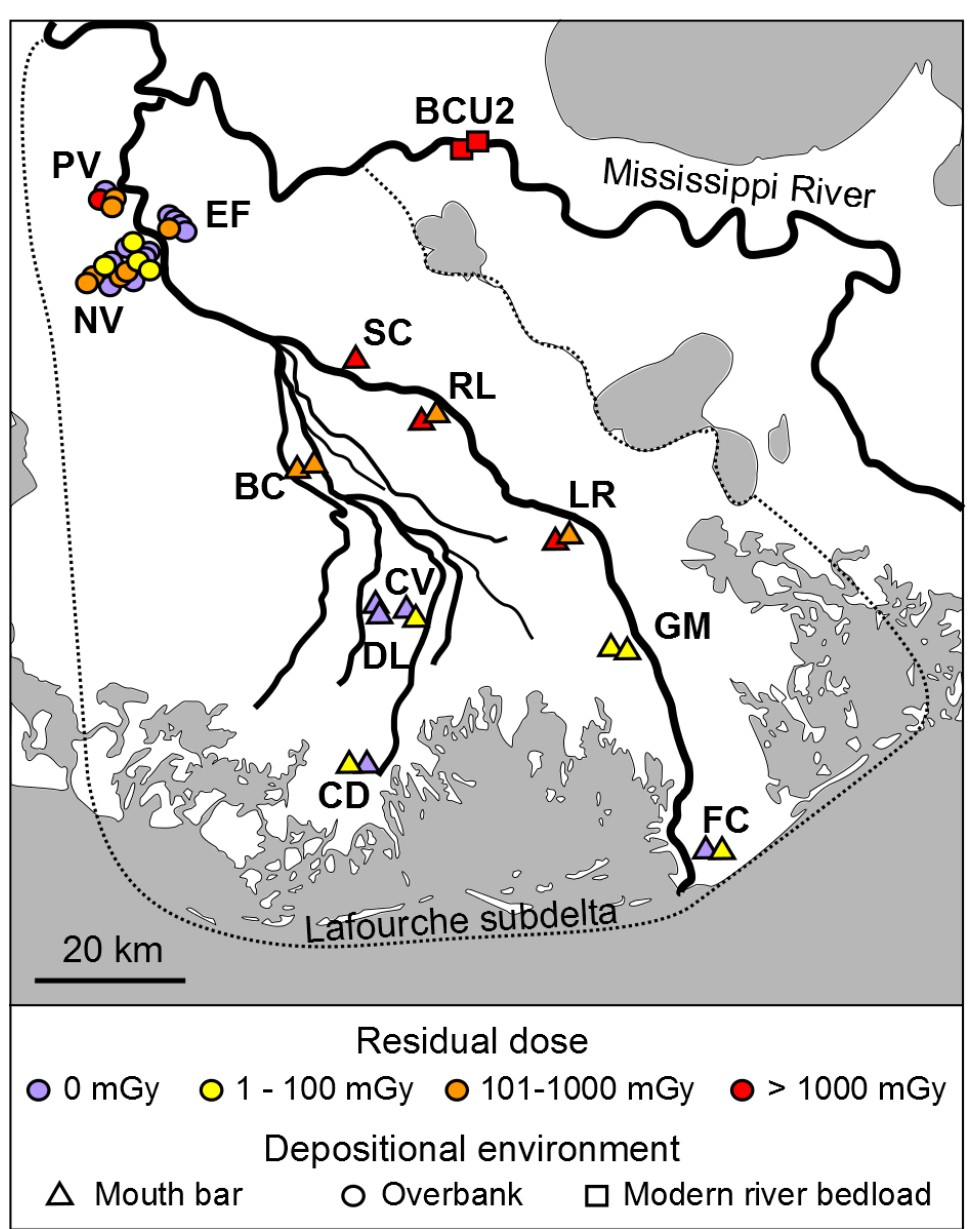


**Figure 8. Geographic distribution of sands and their minimum residual doses, defined as the residual dose minus its uncertainty.**

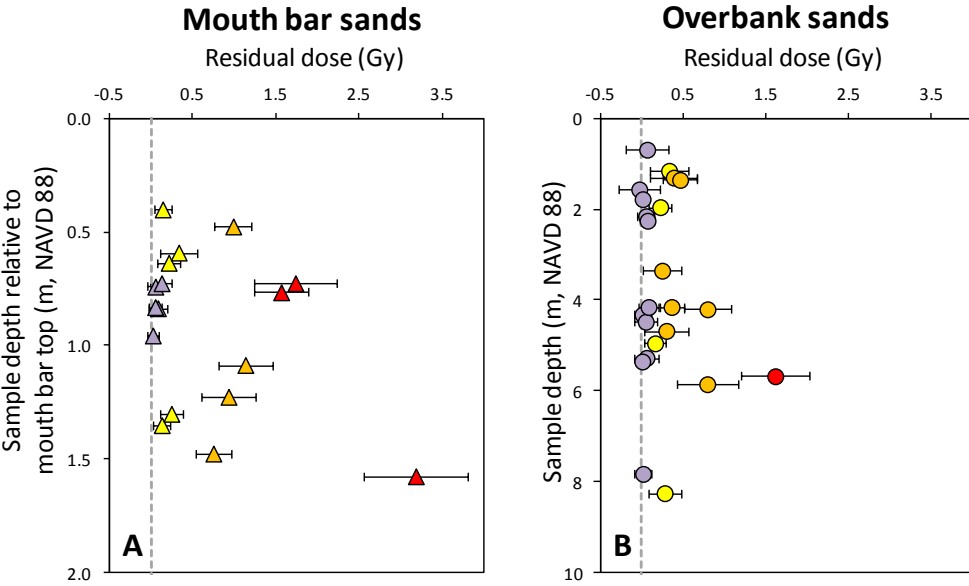

**Figure 9. Bleaching of mouth bar (A) and overbank (B) sands with depth. Mouth bar sand depths are relative to the top of the mouth bar deposit, which formed at roughly sea level. Overbank sand depths are relative to mean sea level. Data points are color-coded by their minimum residual dose (see Fig. 8).**