# Peer review of "Seeking enlightenment of fluvial sediment pathways by OSL signal bleaching of river sediments and deltaic deposits"

_Earth Surface Dynamics, 2018_

## Referee Comment (RC1) · Kreutzer (Referee) · 8 Dec 2018

Dear Elizabeth, Dear Jakob,

Thank you for this exciting and unconventional application of luminescence techniques in Earth Sciences. Your manuscript takes up the challenge to apply luminescence (dating) methods to trace Earth surface processes through signal bleaching trends and mechanisms. As a natural laboratory, you have selected the Mississippi Delta, from where you have reanalysed 49 samples; a dynamic environment and a sufficiently

large sampling size. Given the title of your manuscript, it fits very well into the scope of ESurf, and I am convinced that the topic will foster exciting discussions. The text reads smooth, and the figures are almost well prepared. Nevertheless, given that I was asked to critically review your article, below I will raise some points that, in my view, needs to be addressed before the manuscript can be considered for publication. I will first start with some general remarks, followed by some more detailed comments. Although I am a little bit too late already, maybe the discussion gets extended, and we can have a real debate here.

**General remarks**

- 1. To start with, in the invitation email by the editor, I was asked to draw particular attention to the methodological aspects of your study. Three weeks ago, I printed your manuscript and took it with me into the field where I had no internet access. Unfortunately, I realised too late that I could not correctly review your paper without having access to your article Chamberlain & Wallinga (2018). So far it concerns me, you refer far too often to your article, and your manuscript does not (yet) stand for itself. At some point, I was wondering why you did not merge the two articles. Nevertheless, if the other work (Chamberlain & Wallinga, 2018) is that important, the reader has a right to understand what you have done in there. Your approach is not a standard method, and hence you should adequately describe it. If you feel that it does not fit into your manuscript, you should at least provide sufficient details as a supplement. In particular, since the other article is not open-access and not everyone may have access to it.
- 2. The first sentence was a surprise. You start with "OSL dating [...]". Somehow it does not fit to the title; you submitted your work to ESurf, not to QG. In focus should be your implications for Earth-surface science (and its potential), but not

**ESurfD**
the review of work done in the past by the luminescence dating community. I found in line 75 to 80 a suitable paragraph that I suggest to take as a start. Means, I suggest that you amend the introduction of your manuscript. I don't mean that you have to rewrite it, but a focus shift would considerably improve the attractiveness of the manuscript by starting with 'sediment pathways'

Furthermore, please try to avoid reference dropping. If you talk about significant 'methodological advances' in line with Huntley et al. (1985) and Murray & Wintle (2000), I cannot see Cunningham & Wallinga (2010) or Cunningham & Wallinga (2012) (even not Galbraith et al. 1999) here. I sincerely do not want to diminish the work they have done, but references should be justified and not selectively favour articles in which one of the authors of the here bespoken manuscript is a co-author.

- 3. What I do miss in your manuscript is a proper hypothesis you can test. Maybe you have something in lines 60 to 63, but it is not very obvious. If you reshape your introduction, please try to provide a leading hypothesis you can follow-up and test throughout (please do not hesitate to correct me if I have accidentally overlooked your hypothesis in your text).
- 4. You did already a good job walking the reader through all the results, but still, sometimes I felt abandoned, and I had to look back to the title and try to connect, e.g., the discussion, with the title. The mix of grain size classes, statistical approaches and depositional features makes it not always easy to stay on track. Moreover, I guess the problem is that with the presented data basically every interpretation you want to favour is possible. Locally different transport histories, as you have written it to justify the need for a large dataset, allow different interpretations. Anyway, again, I guess you did a good job, but maybe you can add a subsection (part of the discussion) that honestly identifies, in brief, the limits of your study.

**ESurfD**
- 5. The conclusion feels a little bit off from the rest of the text. I suggest that you rephrase the conclusion to re-connect it with the rest of the manuscript.
- 6. I suggest modifying the title a little bit: "OSL signal" or "luminescence signal" and the word 'Mississippi' should be part of the title.

**1 Detailed comments**

- 1.1 Main text
  - 1. Line 42–49: Perhaps this falls a little bit too short; the effectiveness is also a question of the time domain. You may want to add another sentence clarifying how the cited references have addressed this question.
  - 2. Line 139: Please define 'small aliquots' in brackets.
  - 3. Line 145: According to Rex Galbraith (personal. comm.) it should be termed 'early light' and not 'early background'. I tend to agree, since you subtract signal + background from the signal and not only background.
  - 4. Line 146: Please exchange the reference by Ballarini et al (2007); except I did overlook something here (?)
  - 5. Line 171: What do you mean with 'full details'? I did check you manuscript and the supplement, but 'full details' are somehow missing.
  - 6. Line 178: Please make an estimate of the grain number range.
  - 7. Line 182: 'bin files', please change to 'BIN/BINX-files' and explain what do you mean. I know it, but I doubt that a lot of readers of ESurf know it as well.

**ESurfD**
- 8. Line 184: It is a little bit difficult to evaluate from the data given whether your early-light subtraction is justified at all (please see comments on the supplement where I asked to provide (some) raw data).
- 9. Line 185: Please check the reference, it should be Ballarini et al (2007) (or both, you know it better)
- 10. Line 189: I would not use the term 'age modelling' for calculating a 'mean  $\pm$  standard error'. Besides, the reader needs a leap of faith and trust that everything that you have done here is justified. Please add additional details (in the supplement if required and if better placed).
- 11. Line 191: Please specify the type of standard error. If it is the standard error of the mean, it should be written. If not, you can leave it as it is.
- 12. Line 197-198: I finally double checked your article after could access it. What I do not understand is the estimation of  $\sigma_b$ . Your first step is the application of the CAM to obtain  $\sigma_b$  values for each sample; then you treat the distribution of the obtained overdispersion values with the bootMAM. However, where do you get your  $\sigma_b$  from that you feed into the CAM? Depending on what you put in here, you get everything that you want out from the CAM. Maybe you can clarify this point.
- 13. Line 210-213: Please add a few more information on how you calculated your ages. Since you have the values, please add them to the table in the supplement. Did you use, e.g., *DRAC* to re-calculate your ages or some other software? Please provide a proper reference to make your data analysis conclusive.
- 14. Line 221: Please add corresponding sample numbers in brackets for the residual dose.
- 15. Line 222: What you report here appears to be the (unweighted) mean  $\pm$  the standard deviation; please clarify.
18. Line 241–242: Now it gets a little bit tricky. The way you have written, it implies that you can determine the degree of bleaching also for the silt fraction, but you have only an averaged value. I am convinced that you are aware of it, so please make it clear to the reader.

16. Line 223-224: '[...] plus other methods described in [...]' is a little bit vague;

17. Line 238: Given the uncertainties, I am not sure whether you can make such a statement; also the number of aliguots (max. 4 per sample) is not very large.

- 19. Line 251: Why do you believe that the Pleistocene deposits have suffered from a limited light exposure?
- 20. Line 296: Sorry, I am a little bit lost, I thought for the silt fraction you used only the 'mean  $\pm$  standard error' (cf. line 191), now you state that you have used the CAM(?). Please clarify.
- 21. Line 282: 'improvement with time', please clarify

please detail.

- 22. Line 299: Minor detail, please use 'ka' or 'a' instead of 'years' as you does in the figures/.
- 23. Line 300–309: Readers without a background in luminescence dating will have difficulties to follow you through. I understand that you cannot provide more details, but what does, e.g., 'strong luminescence signals' means to you and why it is important? Details are somehow missing here, but then you talk about a 110 °C TL peak (presumably measured in the UV wavelength range) without further information what did you expect. I suggest that you rephrase this paragraph. I am not even sure whether you need it in the main text. It would be better placed in the supplement.
- 24. Line 308: Why do you talk now about an 'age overestimation; before you did not talk about ages.
- 25. Line 310–315: In light of what you have written in Sec. 3.2 I don't quite get why you apply the CAM on the silt samples. I'm sure I am overlooking something, maybe you can help me out here.
- 26. Line 383: I do agree, the complexity of the dataset and the mixture of grain sizes, statistical (grain size linked) methods and depositional feature makes is extremely hard to follow your conclusions. However, I cannot come-up with a better idea for the moment.
- 27. Line 388: It appears that you somehow mix 'luminescence signal bleaching' with dose and luminescence ages; you should separate the three. Otherwise one may believe that you can draw a similar conclusion from all the three. For example, why you did not focus solely on a (normalised) luminescence signal itself? With this, you would get rid of difficulties regarding your interpretation. For instance, a residual dose of 2 Gy for one sample may indicate the same degree of bleaching as a 3 Gy residual dose for another sample if the dose rate differs. The (total) dose rates (your Table S1) (without accounting for micro-dosimetric effects); did you check for the impact?
- 28. Line 400: "OSL bleaching", should better read "Luminescence signal bleaching". I know it is also your title, but it does not sound right. Another suggestion: "OSL signal bleaching".
- 29. Line 400–404: I do agree with this bullet point but have difficulties to understand why do you believe that this is a direct outcome of your study. I suggest to rephrase this conclusion, please also clarify why your finding only concerns 'sand'.

30. Line 406: 'previous findings': Please add references.

**ESurfD**
31. Line 410: What do you mean with 'there are many unknowns with regard to drivers of luminescence signal bleaching'? The 'drivers' of luminescence signal bleaching is the exposure to light. Please rephrase.

1.2 Figures

- 1. Figure 4: Check: Paleodose' is underlined red?
- 2. Figure 7: Please check red line below 'bootMAM'
- 1.3 Supplement

Maybe you can supplement your manuscript with some BIN/BINX-files that would enable the readers (and the reviewers) to better asses what you have done and play with the dataset and the presented method. I guess it will be exciting to test the impact of different parameter settings. If you don't want to provide access to all data, a selection of a representative dataset would be just fine; as long as your method can be tested independently.

- 1. Please add page numbers to the supplement.
- 2. Figure S1: Please add a proper reference for your data source (e.g., URL + access date).
- 3. Table S1:
  - (a) After I had been carefully looking at your table, I realised that I probably do not understand the reasoning for the quoted minimum residual dose. You basically use the residual dose and subtract the quoted standard error and

**ESurfD**
term it 'minimum'. I would call it 'residual dose' - 'SE(residual dose)', but still, I don't know what you intend to imply, in particular if the value is negative. I suggest to remove this column.

- (b) You have to admit that 'n = 4' or even 'n = 2' is not really a sampling size we should rely on. Not even for 'fine grain'. Do you see any chance to measure a few aliquots more?
- (c) Please round values to meaningful digits.
- (d) Your note under the table: I suggest just 'mean  $\pm$  standard error' not 'central age', in particular since you list dose values, not ages.
- 1. Figure S2: Please add few more information in the figure caption; just "Typical TL response for PV I-4 silt" makes it a little bit hard to see any value in this figure.
- 2. Figure S3: Please remove the red line under 'bootMAM'
- 3. Figure S5: Please add the data source, either to the figure caption or to the map itself and please replace the map by a version with a better resolution; it looks blurred. What means 'NAVD 88 (m)'? The inset is a little bit uninformative (probably readers more familiar with your research area have fewer problems).

Sebastian Kreutzer | Bordeaux, 2018-12-08

**References**

Ballarini, M., Wallinga, J., Wintle, A.G., Bos, A.J.J., 2007. Analysis of equivalent-dose distributions for single grains of quartz from modern deposits. Quaternary Geochronol-
Chamberlain, E.L., Wallinga, J., Shen, Z., 2018. Luminescence age modeling of variably-bleached sediment: Model selection and input. Radiation Measurements 1–7. doi:10.1016/j.radmeas.2018.06.007

Cunningham, A.C., Wallinga, J., 2010. Selection of integration time intervals for quartz OSL decay curves 5, 657–656.

Cunningham, A.C., Wallinga, J., 2012. Realizing the potential of fluvial archives using robust OSL chronologies 12, 98–106. doi:10.1016/j.quageo.2012.05.007

Durcan, J.A., King, G.E., Duller, G.A.T., 2015. DRAC: Dose Rate and Age Calculator for trapped charge dating. Quaternary Geochronology 28, 54–61. doi:10.1016/j.quageo.2015.03.012

Galbraith, R.F., Roberts, R.G., Laslett, G.M., Yoshida, H., Olley, J.M., 1999. Optical dating of single and multiple grains of Quartz from Jinmium Rock Shelter, Northern Australia: Part I, Experimental design and statistical models. Archaeometry 41, 339–364. doi:10.1111/j.1475-4754.1999.tb00987.x

Huntley, D.J., Godfrey-Smith, D.I., Thewalt, M.L.W., 1985. Optical dating of sediments. Nature 313, 105–107.

Murray, A.S., Wintle, A.G., 2000. Luminescence dating of quartz using an improved single-aliquot regenerative-dose protocol. Radiation Measurements 32, 57–73.
**ESurfD**

---

## Referee Comment (RC2) · Cunningham (Referee) · 21 Dec 2018

The manuscript address a very interesting topic, which is the use of OSL data as a proxy for environmental processes. With a range of samples from the Mississippi delta, any inferences on geomorphic processes made from the OSL data will be highly beneficial to geoscience, and relevant to society. The authors use a measure of the residual dose in these samples and observe a dependence on grain size that contrasts with previous studies. They also observe a relationship with the sample age, which implies a change in the transport or sourcing of the sediment over time. These relationships make for an enlightening discussion, but given the complexity of the topic, I have some

reservations about the validity of the results.

There are a number of challenges involved in research of this type, which make it very difficult to draw firm conclusions. The OSL measurements are used to define a 'response' variable, which is the residual dose in this case. The data is then correlated against some potential 'predictor' variables, to try to ascertain which, if any, are driving the changes in the residual dose. However, the predictor variables are also correlated themselves- e.g. the age and distance seaward, so it is not at all obvious which, if any, would be driving the observed changes.

A further complication comes from the definition of residual dose, which is estimated from a statistic of the equivalent dose distributions. This measure is open to error/bias, because it is estimated from imperfect models. For example, the dispersion in the dose distributions is affected by the number of grains in the aliquot; without accounting for this, a spurious correlation of bleaching on grain size might be observed. The authors recognise this effect, and use a measure of bleaching that seeks to account for the differences in aliquot size- the sigma\_b parameter of the minimum age model.

However, there are other reasons that the residual dose statistic might change, even if actual residual dose remains constant. If the aliquot size increases, then the number of well-bleached aliquots is reduced (the 'p' parameter of the minimum age model). We would wish that model performance does not depend on 'p', but it is very likely that is does in some way – my guess is that the results would get more erratic when p is small, with a bias introduced to the burial dose estimate. Another question is in the performance of the models as the burial dose increases: is it possible that the accuracy of burial-dose estimate depends on the actual burial dose? I guess that it would, because at low doses the precision of Des is correlated with their central estimate, and there is an order-of-magnitude difference in the burial doses across the samples being considered.

The observed dependence of residual dose on the sample age, and distance downstream, could both be due to a dependence on the burial dose– which could plausibly arise as an artefact of the data analysis. Before reaching for geomorphic explanations, some effort should be put into checking the validity of the results, and ruling out more mundane explanations for the trends. I can think of a few ways this might be done..

-A simulation of the process. Simulate a poorly bleached distribution, and record how the residual dose statistic depends on the burial dose. Is there a dependency on the number of grains, or the sensitivity distribution? What conditions would be necessary to induce a dependence on De?

-Experimental simulation. Create artificial poorly-bleached populations by giving different beta doses to different aliquots. Mix the aliquots in relevant proportions, measure De and calculate the bleaching statistic. Is there trend with 'burial' dose?

-Apply the method on an unrelated dataset. This is probably the easiest test. Take similar data from elsewhere –the Rhine-Meuse for example– and repeat the analysis. If you see a similar trend with De/age, then it is probably an artefact.

An addition along these lines would greatly strengthen the manuscript, by permitting more confidence in the validity of the conclusions. In addition, I would hope to see some recognition of the limitations of the methods used: a description of the key assumptions in the methods section, and discussion section that reflects on the validity of the results, given the assumptions and limitations.

**other points**

Title – may need a softer title that reflects the caveats above. 'Seeking enlightenment of sediment pathways...'?

192 and elsewhere. - There is a difference in method used for modern and palaeo data. Is this necessary? Could not the unlogged mam3 be used for the moderen data?

195 - there needs to be a reasonable explanation of the method being used to evaluate the residual dose. This should be a good paragraph, enough for an informed reader

СЗ

to understand the approach without looking up your earlier paper. It is also important that the key assumptions and limitations of the method are described; in the discussion section, the results should be interpreted with regard to these limitations.

219 - 'channel depth'.. or rather, the sampling depth within the channel?

255 – The residual dose is defined after subtracting the 1-sigma uncertainty. This seems odd, and I couldn't find an explanation in the previous paper.

265 and fig. 5 – the relationship with grain size looks impressive, but there are other possible explanations for the relationship. For instance, it seems the smaller grain sizes relate exclusively to modern sediments, while the larger grain sizes relate to holocene deposits (this is the problem of correlated variables again). Then there is the question of how well the statistic performs when the non-bleaching parameters change- number of grains, or the sensitivity distribution.

section 4.6 - I found this section a bit confusing. The objective is to compare the bleaching of sand and silt fractions of the same sample, is it not? But data is plotted as ages, not doses (or residuals), and using different age models for the different fractions. There are very few samples, and eventually it is suggested that they are well bleached anyway. It might be best to omit this section entirely, as it doesn't seem to add anything to the paper.

---

## Author Comment (AC1) · 7 Jan 2019

Dear Sebastian and Alastair,

Thank you for your insightful and careful reviews of our work. We are pleased to hear that you find our work exciting and of interest to Esurf readership, with implications for science and society.

We have considered all your comments, and we have improved the manuscript in response.

Please find attached a detailed response to each of your comments (referred to be-

low as the supplement to the comment). We are prepared to submit an improved manuscript that incorporates the outlined changes.

Again, many thanks.

Liz Chamberlain, on behalf of all authors

Please also note the supplement to this comment:
https://www.earth-surf-dynam-discuss.net/esurf-2018-76/esurf-2018-76-AC1-supplement.pdf

---

## Author Response (AR1)

**ESURF-2018-76: Authors' response to reviewer comments**

Elizabeth Chamberlain and Jakob Wallinga

**General reply**

Dear Andreas,

Thank you for organizing this efficient review process and sending our work to two highly-qualified reviewers. We also thank the reviewers for helping us to improve the manuscript.

Overall, both reviewers judged our manuscript to be exciting and of interest to the Esurf readership. The writing and presentation of our science seemed to be well received -- the quality of the reviewer comments demonstrate that they understood the key aspects our research, the suggested edits regarding the writing style were moderate, and those regarding figure presentation were very minor. We have made modifications where we felt they improved the work, as detailed in the Replies to Reviews.

In addition to the positive response, both reviewers also noted that this study employs unconventional methods, introduced and tested in a separate article by our team that is in press at *Radiation Measurements*. This article has been available online since June 4, 2018 at https://www.sciencedirect.com/science/article/abs/pii/S135044871730776X, and is part of the LED proceedings special issue.

The reviewers asked that we provide a summary of those methods here, so that the present article may stand alone. In response, we have added paragraphs to section 3.2 describing 1) our novel sigma_b estimation approach, and 2) our approach to residual dose estimation for each sediment type.

We have also added text in the methods sections, outlining the assumptions of the methods, and a paragraph in section 5.1, discussing the limitations of our research. These changes were requested by both reviewers. Please note that we have also re-ordered the figures in the supplement to match the order they are referenced in the main text. The reviewers requested changes to the title; in response the manuscript is now entitled "Seeking enlightenment of fluvial sediment pathways by OSL signal bleaching of river sediments and deltaic deposits".

Sebastian Kreutzer (Reviewer 1) advised that we better frame our introduction with regard to sediment fingerprinting. In response, we have reworded the introduction. Alastair Cunningham (Reviewer 2) asked for additional consideration on our age modeling. In response, we have added new data and a figure to the supplement (Table S1, Fig. S2) plus supporting text in section 3.2.

In addition to these significant improvements, please see other changes and/or responses detailed in our Replies to Reviewers.

Kind regards,

Liz Chamberlain, on behalf of all authors.

**Replies to Reviews: Review 1**

Dear Elizabeth,

Dear Jakob,

Thank you for this exciting and unconventional application of luminescence techniques in Earth Sciences. Your manuscript takes up the challenge to apply luminescence (dating) methods to trace Earth surface processes through signal bleaching trends and mechanisms. As a natural laboratory, you have selected the Mississippi Delta, from where you have reanalysed 49 samples; a dynamic environment and a sufficiently large sampling size. Given the title of your manuscript, it fits very well into the scope of ESurf, and I am convinced that the topic will foster exciting discussions. The text reads smooth, and the figures are almost well prepared. Nevertheless, given that I was asked to critically review your article, below I will raise some points that, in my view, needs to be addressed before the manuscript can be considered for publication. I will first start with some general remarks, followed by some more detailed comments. Although I am a little bit too late already, maybe the discussion gets extended, and we can have a real debate here.

Dear Sebastian,

   Thank you for your thoughtful and detailed review of our work. We are pleased to hear that you find our study exciting and suitable for the scope of *Esurf*. Please see our response to your general and specific comments below.

*General Remarks:*

1. To start with, in the invitation email by the editor, I was asked to draw particular attention to the methodological aspects of your study. Three weeks ago, I printed your manuscript and took it with me into the field where I had no internet access. Unfortunately, I realised too late that I could not correctly review your paper without having access to your article Chamberlain & Wallinga (2018). So far it concerns me, you refer far too often to your article, and your manuscript does not (yet) stand for itself. At some point, I was wondering why you did not merge the two articles. Nevertheless, if the other work (Chamberlain & Wallinga, 2018) is that important, the reader has a right to understand what you have done in there. Your approach is not a standard method, and hence you should adequately describe it. If you feel that it does not fit into your manuscript, you should at least provide sufficient details as a supplement. In particular, since the other article is not open-access and not everyone may have access to it.

We apologize that the prior paper describing the methods employed here was not accessible to you while you were conducting the review, due to a lack of internet access. We imagine that made some aspects of the review rather difficult, and we agree that this experience highlights the need for new text briefly summarizing the methods presented in Chamberlain et al. (in press). We have addressed this by adding text outlining the steps for sigma_b calculation and adding paragraphs about residual dose estimation, both in section 3.2.

Like you, we also questioned whether the two studies would be better published together or separate. We ultimately decided to publish the concepts as separate manuscripts for two primary reasons: 1) The scope of our publication in *Radiation Measurements* is very different from the scope of the present manuscript for *Esurf*. The former deals with the technical details of luminescence statistics and age modeling, while the latter is an applied manuscript that uses luminescence to test Earth surface processes. As such, we judged that the two would appeal to very different audiences, and we did not want to lose either audience, for example by overwhelming fluvial researchers with OSL-specific terminology or by providing excessive detail on fluvial systems which may not be of interest to all OSL specialists. 2) The two papers are too lengthy and dense to be combined as a single manuscript. As you note in General Comment 4, the dataset herein is already quite complex. Furthermore, the two manuscripts in sum contain 16 essential figures plus 9 figures in their supplements, as well as numerous tables. They also yield very different sets of conclusions, which speak to different aspects of the science of luminescence dating and its uses. In summary, we judged the two concepts to be full and independent stories with different key points and readership, meriting presentation as separate papers.

2. The first sentence was a surprise. You start with "OSL dating [...]". Somehow it does not fit to the title; you submitted your work to ESurf, not to QG. In focus should be your implications for Earth-surface science (and its potential), but not the review of work done in the past by the luminescence dating community. I found in line 75 to 80 a suitable paragraph that I suggest to take as a start. Means, I suggest that you amend the introduction of your manuscript. I don't mean that you have to rewrite it, but a focus shift would considerably improve the attractiveness of the manuscript by starting with 'sediment pathways'.

We agree, and we have reworked the introduction to fit the title and aims of the manuscript.

3. What I do miss in your manuscript is a proper hypothesis you can test. Maybe you have something in lines 60 to 63, but it is not very obvious. If you reshape your introduction, please try to provide a leading hypothesis you can follow-up and test throughout (please do not hesitate to correct me if I have accidentally overlooked your hypothesis in your text).

Given the novelty of our approach, we had no clear expectations beforehand, and did not work with a hypothesis to be tested. This is explorative research, which does in our mind not benefit from working with a hypothesis. Yet, we agree that the aim could be phrased and explained more

clearly, and we did so by modifying a sentence toward the end of the introduction stating "All combined, these data allow us to test whether OSL signal bleaching varies across time, space, grain size, and depositional environment, even within a single fluviodeltaic system." This is revisited in the discussion.

4. You did already a good job walking the reader through all the results, but still, sometimes I felt abandoned, and I had to look back to the title and try to connect, e.g., the discussion, with the title. The mix of grain size classes, statistical approaches and depositional features makes it not always easy to stay on track. Moreover, I guess the problem is that with the presented data basically every interpretation you want to favour is possible. Locally different transport histories, as you have written it to justify the need for a large dataset, allow different interpretations. Anyway, again, I guess you did a good job, but maybe you can add a subsection (part of the discussion) that honestly identifies, in brief, the limits of your study.

We agree that it is quite a lot of co-mingled data (this is in part why the statistical methods are detailed in a separate paper). We also agree that the data are not straightforward, given their complexity, and we added a paragraph in section 5.1 discussing the limitations of our study.

5. The conclusion feels a little bit off from the rest of the text. I suggest that you rephrase the conclusion to re-connect it with the rest of the manuscript.

We added lead-in sentences to connect the conclusions bullet points to the manuscript.

6. I suggest modifying the title a little bit: "OSL signal" or "luminescence signal" and the word 'Mississippi' should be part of the title.

We agree that replacing "OSL" with "OSL signal" in the title is an improvement, and we adopted this suggestion. We prefer not to add Mississippi to the title; although it is indeed the natural laboratory we use to explore applicability of our approach, the focus of our paper is on the approach rather than the location. With our new title 'seeking enlightenment', following a suggestion from the other reviewer, we feel that this focus is now clear.

*Detailed comments*

1. Line 42–49: Perhaps this falls a little bit too short; the effectiveness is also a question of the time domain. You may want to add another sentence clarifying how the cited references have addressed this question.

We added "This has been approached through tests of modern sediments or those of independently-constrained depositional ages".

2. Line 139: Please define 'small aliquots' in brackets.

We added "(that is, numerous subsamples for each sample, each containing ~23-108 grains)".

3. Line 145: According to Rex Galbraith (personal. comm.) it should be termed 'early light' and not 'early background'. I tend to agree, since you subtract signal + background from the signal and not only background.

We can see the value in the term "early light", however we decline to change our terminology here, as "early background" is widely used and understood, and is consistent with the reference we use here.

4. Line 146: Please exchange the reference by Ballarini et al (2007); except I did overlook something here (?)

Cunningham & Wallinga (2010) is a suitable reference here because they tested the ideal integration intervals, which we apply.

5. Line 171: What do you mean with 'full details'? I did check you manuscript and the supplement, but 'full details' are somehow missing.

Munoz et al. (2018) did not specify certain aspects such as the collection methods or some details of the luminescence analysis (e.g., the interval of background integration). We give those details here, and think we now provided all essential information. We have changed "full details" to "essential details".

6. Line 178: Please make an estimate of the grain number range.

We added these values.

7. Line 182: 'bin files', please change to 'BIN/BINX-files' and explain what do you mean. I know it, but I doubt that a lot of readers of ESurf know it as well.

We changed to "BIX/BINX-files, generated through luminescence measurement using Risø readers".

8. Line 184: It is a little bit difficult to evaluate from the data given whether your early-light subtraction is justified at all (please see comments on the supplement where I asked to provide (some) raw data).

Early background subtraction is generally a good practice on fluvial sediment, especially when poor bleaching is suspected. EBS is especially a reasonable choice here because, as we state in the text, EBS was found to yield younger and more accurate ages for Mississippi delta deposits of historically-known depositional ages (Shen and Mauz, 2012).

9. Line 185: Please check the reference, it should be Ballarini et al (2007) (or both, you know it better).

As we state above, Cunningham & Wallinga (2010) is a suitable reference here because they tested the ideal integration intervals, which we apply.

10. Line 189: I would not use the term 'age modelling' for calculating a 'mean _ standard error'. Besides, the reader needs a leap of faith and trust that everything that you have done here is justified. Please add additional details (in the supplement if required and if better placed).

We prefer "age modeling" here because it is the most correct term to our knowledge; "paleodose determination" is not valid for our application of the mean or the CAM to determine the average (and thereby, residual) dose on the sample, and we are unaware of a more fitting term that encompasses the number of statistical treatments we applied.

Our decisions regarding age modeling may seem to be a leap of faith, however, we very carefully tested across multiple models and made informed decisions prior to framing our data for the manuscript. We added a column to Table S1 that shows the $D_{e,MEAN}$ we calculated for samples that were also modeled with the CAM, and a new figure (Fig. S2) that allows for a visual comparison; this way the reader can better judge our choice of statistical models. These additional data show that the CAM and a mean yield similar central doses; we chose to feature the CAM in the primary text because this is one of the most widely used age models, and therefore our assessment is generic and relevant to other studies. The CAM is not suitable for very young/modern samples, because we have previously observed that it preferentially weights the higher dose aliquots (due to lower relative uncertainty), and thus risks overestimating the central dose of young, well-bleached sediments. For this reason, we used a mean on the modern river silt.

For consistency, we now use a mean and standard error (rather than the CAM) on Late Holocene silts. We also present the CAM doses (Table S2), which are nearly identical to those obtained with the mean and standard error.

11. Line 191: Please specify the type of standard error. If it is the standard error of the mean, it should be written. If not, you can leave it as it is.

It is the standard deviation of the $D_e$s divided by the square root of $n$.

12. Line 197-198: I finally double checked your article after could access it. What I do not understand is the estimation of _b. Your first step is the application of the CAM to obtain _b values for each sample; then you treat the distribution of the obtained overdispersion values with the bootMAM. However, where do you get your _b from that you feed into the CAM? Depending on what you put in here, you get everything that you want out from the CAM. Maybe you can clarify this point.

We do not feed a sigma_b to the CAM. We feed a sigma_b of [0 0] to the bootMAM, when modeling the overdispersion of the dataset. This is now described in section 3.2..

13. Line 210-213: Please add a few more information on how you calculated your ages. Since you have the values, please add them to the table in the supplement. Did you use, e.g., DRAC to re-calculate your ages or some other software? Please provide a proper reference to make your data analysis conclusive.

Ages for the sand-silt pairs are provided in Table S2. We did not calculate ages for the other samples; rather, we calculated the average residual dose for various groups and obtained an average age by dividing this by the average dose rate, described in section 3.3.

We used a conventional excel sheet with standard inputs to determine dose rates; we do not think this detail is needed in the manuscript.

We added a sentence in section 3.3 stating that the sand-silt pair ages were "...calculated by dividing the paleodose of each sample by its dose rate, and propagating uncertainties in quadrature."

14. Line 221: Please add corresponding sample numbers in brackets for the residual dose.

This is a range of values for all samples within the group -- as stated, specific values for each sample can be found in Table S1.

15. Line 222: What you report here appears to be the (unweighted) mean _ the standard deviation; please clarify.

Yes, this is an unweighted mean and standard error. We clarified in section 3.2.

16. Line 223-224: '[...] plus other methods described in [...]' is a little bit vague; please detail.

This relates to things like fitting of the dose-response curve through the origin, and aliquot acceptance criteria. Such details are outside of the scope of this manuscript but interested readers can find them in the cited paper.

17. Line 238: Given the uncertainties, I am not sure whether you can make such a statement; also the number of aliquots (max. 4 per sample) is not very large.

We clearly state that this is one of two possible interpretations, which allows the reader to judge what they consider likely. While n=4 is a small aliquot sample size, a silt aliquot can contain more than 1 million grains, meaning that the measurement of even one silt aliquot yields more averaged signals than 100+ small-diameter sand aliquots may provide. This is evidenced by the low standard error of the silt samples, even on the small sample sizes.

18. Line 241–242: Now it gets a little bit tricky. The way you have written, it implies that you can determine the degree of bleaching also for the silt fraction, but you have only an averaged value. I am convinced that you are aware of it, so please make it clear to the reader.

We can, in fact, estimate bleaching of these silt samples, because they are modern analogues and should have a zero dose if well bleached. We added text to section 3.2 to clarify how we judge bleaching (residual doses) for different sediment groups.

19. Line 251: Why do you believe that the Pleistocene deposits have suffered from a limited light exposure?

The estimated residual age indicates that these grains are older than Holocene-aged. That they retain a residual dose greater than 25 Gy indicates they have not been fully reset during river transport. However, they may be older than Pleistocene-aged...we now use the term "pre-Holocene" rather than Pleistocene.

20. Line 296: Sorry, I am a little bit lost, I thought for the silt fraction you used only the 'mean _ standard error' (cf. line 191), now you state that you have used the CAM(?). Please clarify.

We agree that this is confusing, and we now present the paleo-deposit silt doses (and ages) calculated with a mean and standard error (please see our reply to your comment on Line 189). We now clarify this in section 3.2.

21. Line 282: 'improvement with time', please clarify

We changed "improvement" to "better bleaching".

22. Line 299: Minor detail, please use 'ka' or 'a' instead of 'years' as you does in the figures/.

I don't disagree that your suggestion would be technically correct, but to me it doesn't read smoothly. We will leave it to the editor to decide.

23. Line 300–309: Readers without a background in luminescence dating will have difficulties to follow you through. I understand that you cannot provide more details, but what does, e.g., 'strong luminescence signals' means to you and why it is important? Details are somehow missing here, but then you talk about a 110 _C TL peak (presumably measured in the UV wavelength range) without further information what did you expect. I suggest that you rephrase this paragraph. I am not even sure whether you need it in the main text. It would be better placed in the supplement.

We agree, and have moved these OSL-specific details to the supplement.

24. Line 308: Why do you talk now about an 'age overestimation; before you did not talk about ages.

For the paleo-silt samples we test ages rather than residual doses because 1) residual doses can't be calculated as the $D_{e,CAM}$ - $D_{e,bootMAM}$ due to the high averaging of signals within aliquots, and 2) dose rates experienced by sand and silt grains vary, even within the same bulk sediment matrix. We now discuss this in section 3.2.

25. Line 310–315: In light of what you have written in Sec. 3.2 I don't quite get why you apply the CAM on the silt samples. I'm sure I am overlooking something, maybe you can help me out here.

There is no real harm in using CAM on these samples as it produces similar doses as the mean (see Table S2), however, we agree that this is confusing. We now use a mean and standard error on all silt samples. Doses obtained with both approaches are given in the supplement (Table S2).

26. Line 383: I do agree, the complexity of the dataset and the mixture of grain sizes, statistical (grain size linked) methods and depositional feature makes is extremely hard to follow your conclusions. However, I cannot come-up with a better idea for the moment.

We added a paragraph immediately before this, that acknowledges the complexity of the dataset and thus the limitations of interpreting it.

27. Line 388: It appears that you somehow mix 'luminescence signal bleaching' with dose and luminescence ages; you should separate the three. Otherwise one may believe that you can draw a similar conclusion from all the three. For example, why you did not focus solely on a (normalised) luminescence signal itself? With this, you would get rid of difficulties regarding your interpretation. For instance, a residual dose of 2 Gy for one sample may indicate the same degree of bleaching as a 3 Gy residual dose for another sample if the dose rate differs. The (total) dose rates (your Table S1) (without accounting for micro-dosimetric effects); did you check for the impact?

In our mind, both remnant dose and remnant age are of importance. The remnant dose upon deposition is the most direct measurement of the degree of bleaching, and thus relevant when investigation the dependency of bleaching on depositional context or sample properties. Yet, when dating fluvial samples, one is concerned about potential age overestimation due to incomplete bleaching. Therefore the 'remnant age', as inferred from the remnant dose in combination with the sample dose rate, is also relevant.

28. Line 400: "OSL bleaching", should better read "Luminescence signal bleaching". I know it is also your title, but it does not sound right. Another suggestion: "OSL signal bleaching".

We have changed to "OSL signal" bleaching in the text and title, as we agree this is more correct.

29. Line 400–404: I do agree with this bullet point but have difficulties to understand why do you believe that this is a direct outcome of your study. I suggest to rephrase this conclusion, please also clarify why your finding only concerns 'sand'.

This is drawn from the unexpected trend we identify in mouth bar sand residual doses, and is described in detail in section 5.1.

30. Line 406: 'previous findings': Please add references.

References do not belong in the bullet point conclusions; these studies are discussed earlier in the text.

31. Line 410: What do you mean with 'there are many unknowns with regard to drivers of luminescence signal bleaching'? The 'drivers' of luminescence signal bleaching is the exposure to light. Please rephrase.

We rephrased to "the processes that drive..."

*1.2 Figures*

1. Figure 4: Check: Paleodose' is underlined red?

This seems to be an issue with the conversion to PDF, which we will mind when we upload a revised version.

2. Figure 7: Please check red line below 'bootMAM'

Same as above, an issue with PDF conversion.

*1.3 Supplement*

Maybe you can supplement your manuscript with some BIN/BINX-files that would enable the readers (and the reviewers) to better asses what you have done and play with the dataset and the presented method. I guess it will be exciting to test the impact of different parameter settings. If you don't want to provide access to all data, a selection of a representative dataset would be just fine; as long as your method can be tested independently.

Although we warmly support full access to data, this is unfortunately not possible for these data at this point in time. Reasons are that the data is not sole property of the authors, and that follow-on publications are planned using part of the same dataset. Publishing the data at this point would jeopardize these publications. Yet, the methods are fully documented in our papers, and can be applied to other large datasets. If problems occur, we would be happy to assist and support.

1. Please add page numbers to the supplement.

Done.

2. Figure S1: Please add a proper reference for your data source (e.g., URL + access date).

Done.

3. Table S1:

(a) After I had been carefully looking at your table, I realised that I probably do not understand the reasoning for the quoted minimum residual dose. You basically use the residual dose and subtract the quoted standard error andterm it 'minimum'. I would call it 'residual dose' - 'SE(residual dose)', but still, I don't know what you intend to imply, in particular if the value is negative. I suggest to remove this column.

We need a way of accounting for uncertainty on the residual dose, for classifying it (e.g., Fig. 8), and deemed this the best approach. We added text in section 4.2 to better describe this approach and its limitation (that some samples classified as well bleached may in fact possess small residual doses).

(b) You have to admit that 'n = 4' or even 'n = 2' is not really a sampling size we should rely on. Not even for 'fine grain'. Do you see any chance to measure a few aliquots more?

We generally like to measure at least 6 aliquots per silt sample, however, the material was very limited. Still, a silt aliquot can contain over 1 million grains, so the measurement of even one aliquot yields more averaged signals than 100+ small-diameter sand aliquots may provide. This is evidenced by the low standard error of the silt samples, even on the very small sample sizes.

(c) Please round values to meaningful digits.

Done.

(d) Your note under the table: I suggest just 'mean _ standard error' not 'central age', in particular since you list dose values, not ages.

We have removed this text, in accordance with changes to the table.

1. Figure S2: Please add few more information in the figure caption; just "Typical TL response for PV I-4 silt" makes it a little bit hard to see any value in this figure.

Following your suggestion on Lines 300-309, we moved details of our feldspar-contamination interpretation from the main text to this caption.

2. Figure S3: Please remove the red line under 'bootMAM'

This seems to also be an artefact of the pdf conversion; we will mind it in the future.

3. Figure S5: Please add the data source, either to the figure caption or to the map itself and please replace the map by a version with a better resolution; it looks blurred. What means 'NAVD 88 (m)'? The inset is a little bit uninformative (probably readers more familiar with your research area have fewer problems).

This is the best resolution version that we have, and we think it is legible for the supplement. NAVD 88 is the North American Vertical Datum of 1988, a standard elevation benchmark used in US research.

**Replies to Reviews: Review 2**

The manuscript address a very interesting topic, which is the use of OSL data as a proxy for environmental processes. With a range of samples from the Mississippi delta, any inferences on geomorphic processes made from the OSL data will be highly beneficial to geoscience, and relevant to society. The authors use a measure of the residual dose in these samples and observe a dependence on grain size that contrasts with previous studies. They also observe a relationship with the sample age, which implies a change in the transport or sourcing of the sediment over time. These relationships make for an enlightening discussion, but given the complexity of the topic, I have some reservations about the validity of the results.

There are a number of challenges involved in research of this type, which make it very difficult to draw firm conclusions. The OSL measurements are used to define a 'response' variable, which is the residual dose in this case. The data is then correlated against some potential 'predictor' variables, to try to ascertain which, if any, are driving the changes in the residual dose. However, the predictor variables are also correlated themselves- e.g. the age and distance seaward, so it is not at all obvious which, if any, would be driving the observed changes.

A further complication comes from the definition of residual dose, which is estimated from a statistic of the equivalent dose distributions. This measure is open to error/bias, because it is estimated from imperfect models. For example, the dispersion in the dose distributions is affected by the number of grains in the aliquot; without accounting for this, a spurious correlation of bleaching on grain size might be observed. The authors recognise this effect, and use a measure of bleaching that seeks to account for the differences in aliquot size- the $\sigma_b$ parameter of the minimum age model. However, there are other reasons that the residual dose statistic might change, even if actual residual dose remains constant. If the aliquot size increases, then the number of well-bleached aliquots is reduced (the 'p' parameter of the minimum age model).

We would wish that model performance does not depend on 'p', but it is very likely that is does in some way – my guess is that the results would get more erratic when p is small, with a bias introduced to the burial dose estimate. Another question is in the performance of the models as the burial dose increases: is it possible that the accuracy of burial-dose estimate depends on the actual burial dose? I guess that it would, because at low doses the precision of Des is correlated with their central estimate, and there is an order order-of-magnitude difference in the burial doses across the samples being considered.

The observed dependence of residual dose on the sample age, and distance down- stream, could both be due to a dependence on the burial dose– which could plausibly arise as an artefact of the data analysis. Before reaching for geomorphic explanations, some effort should be put into checking the validity of the results, and ruling out more mundane explanations for the trends. I can think of a few ways this might be done..

–A simulation of the process. Simulate a poorly bleached distribution, and record how the residual dose statistic depends on the burial dose. Is there a dependency on the number of grains, or the sensitivity distribution? What conditions would be necessary to induce a dependence on De?

–Experimental simulation. Create artificial poorly-bleached populations by giving different beta doses to different aliquots. Mix the aliquots in relevant proportions, measure De and calculate the bleaching statistic. Is there trend with 'burial' dose?

–Apply the method on an unrelated dataset. This is probably the easiest test. Take similar data from elsewhere –the Rhine-Meuse for example– and repeat the analysis. If you see a similar trend with De/age, then it is probably an artefact.

An addition along these lines would greatly strengthen the manuscript, by permitting more confidence in the validity of the conclusions. In addition, I would hope to see some recognition of the limitations of the methods used: a description of the key assumptions in the methods section, and discussion section that reflects on the validity of the results, given the assumptions and limitations.

Dear Alastair,

Thank you for your helpful review. We are happy to hear that you find the topic to be of value to science and society.

You describe two primary concerns with correlating OSL bleaching to environmental processes: 1) parsing the different potential explanatory processes ('predictors'), which are often themselves related, and 2) validity of bleaching estimates.

To address your concerns about parsing the different processes that may drive bleaching of riverine sediments, we already indicated that this is hampered by correlation of potential predictors. To highlight this, we added a paragraph in section 5.1 discussing the limitations our study.

Regarding the methodological concerns, we agree that confidence in our methods is important, and that age models can sometimes introduce bias as you describe. We added text to acknowledge the assumptions of our methods, for example in sections 3.2 and 4.2.

You've suggested a few approaches to further testing our method. Although we acknowledge that this would be interesting, we choose not to incorporate this suggestion. Preparing mixed samples is not practically possible for us at this point. Using Rhine-Meuse data would also be problematic, as we would need to compare results from different sites, which would require additional geologic and stratigraphic background. We think this would make the manuscript overly long and convoluted. Simulations would potentially be feasible, but including methods and results of such an exercise would unnecessarily complicate and lengthen the manuscript.

Ultimately, we feel these additional tests are not necessary, because 1) our bootMAM results are in perfect stratigraphical and/or progradational order, are in agreement with radiocarbon age control, and are thus likely to be correct, 2) The CAM ages are similar to the unweighted means, and provide a genuine estimate of the central dose, and 3) If both bootMAM and CAM estimates are robust (see point 1 and 2), the residual dose cannot be a methodological artefact. To support this line of reasoning, we have added the values for $D_{e,MEAN}$ in Table S1, and provided a visual comparison with $D_{e,CAM}$ in Fig. S2. We explicitly address this issue through new text in section 3.2.

We also wish to point out that we did not change the aliquot size, however, the number of grains per aliquot did vary among grain sizes. We are confident that the strongly-vetted use of tailored sigma_b values accommodates the different numbers of grains per disk in a way that makes comparison across grain sizes valid, and we tested this in Chamberlain et al. (in press).

*Other points*

Title – may need a softer title that reflects the caveats above. 'Seeking enlightenment of sediment pathways: : :' ?

We welcome this suggestion and have modified the title to "Seeking enlightenment of fluvial sediment pathways by OSL signal bleaching of river sediments and deltaic deposits".

192 and elsewhere. - There is a difference in method used for modern and palaeo data. Is this necessary? Could not the unlogged mam3 be used for the moderen data?

It is true that we use different methods to judge the residual doses of sand isolated from the modern river sediment vs. paleo-deposits. This is now justified in section 3.2. Modern river sand residual doses are determined by $D_{e,CAM}$, because $D_{e,CAM}$ should be zero if all grains are well bleached. This is the most simple way to accurately judge bleaching of these sediments, and therefore the most valid. Furthermore, the bootMAM did not yield a zero dose for the coarsest grain size of the modern river sand, suggesting that assessing bleaching of this fraction as $D_{e,CAM}$ - $D_{e,bootMAM}$ (as we do for paleo-deposits) would not yield a valid estimate. In our mind, this does not invalidate using $D_{e,bootMAM}$ as the paleodose estimate for the paleo-deposits; as we highlight in the manuscript, the channel bottom sediments in transit in the modern Mississippi channel are

likely to be less well bleached than the paleo-deposits sampled from the Bayou Lafourche system.

By contrast, paleo-deposit residual doses are judged as $D_{e,CAM}$ - $D_{e,bootMAM}$ . We believe this is a valid approach because the bootMAM yielded ages that are stratigraphically consistent and in line with radiocarbon constraints.

In other words, the methods are slightly different because the information at hand for the two sediment groups (modern vs paleo) is different. We chose to use the most valid method possible for each group, given what we knew about the sediments.

195 - there needs to be a reasonable explanation of the method being used to evaluate the residual dose. This should be a good paragraph, enough for an informed reader to understand the approach without looking up your earlier paper. It is also important that the key assumptions and limitations of the method are described; in the discussion section, the results should be interpreted with regard to these limitations.

We agree, and we added paragraphs explaining residual dose estimation and its assumptions in section 3.2.

219 – 'channel depth'.. or rather, the sampling depth within the channel?

We changed to "sampling depth below the water surface".

255 – The residual dose is defined after subtracting the 1-sigma uncertainty. This seems odd, and I couldn't find an explanation in the previous paper.

We use this approach because we need a way of accounting for uncertainty on the residual dose, for classifying it (e.g., Fig. 8). We added a justification and acknowledgement of this limitation in section 4.2. " As there is uncertainty on the $D_{e,CAM}$ and $D_{e,MAM}$ values from which residual doses were calculated, there is also uncertainty on the residual doses. The bleaching of each Late Holocene-aged sample was classified by its minimum residual dose, defined as the residual dose minus 1σ uncertainty. This means that some samples classified as well bleached may have possessed small residual doses."

265 and fig. 5 – the relationship with grain size looks impressive, but there are other possible explanations for the relationship. For instance, it seems the smaller grain sizes relate exclusively to modern sediments, while the larger grain sizes relate to holocene deposits (this is the problem of correlated variables again). Then there is the question of how well the statistic performs when the non-bleaching parameters change- number of grains, or the sensitivity distribution.

We agree that correlated variables is an issue in the interpretation of our data, and we address this limitation in a new paragraph in section 5.1. While the fine grain data in Fig. 5 are from only modern samples, we also observe sufficient bleaching of older fine grain deposits through tests

of the sand-silt pairs. We are confident that we correct well for the non-bleaching parameters of the statistics through the use of our adapted sigma_b values.

section 4.6 – I found this section a bit confusing. The objective is to compare the bleaching of sand and silt fractions of the same sample, is it not? But data is plotted as ages, not doses (or residuals), and using different age models for the different fractions. There are very few samples, and eventually it is suggested that they are well bleached anyway. It might be best to omit this section entirely, as it doesn't seem to add anything to the paper.

Yes, the idea is to assess bleaching of sand and silt isolated from the same sample, so that we can make some inferences about bleaching of silt of paleodeposits. We do not wish to omit this, as we believe it relates to our testing of modern river silt and is important for judging bleaching of fine grains across time (modern vs. paleo). Instead, we now better explain this test through new text in section 3.2, and we improved the presentation of the results by adding a second panel to Fig. 7, which allows the reader to better judge bleaching between the two grain size fractions (Fig. 7B).

Testing the bleaching of modern river silt is fairly straightforward, because any dose greater than zero indicates incomplete resetting, and the residual dose can be determined by a mean. However, a similar test is not possible on silts isolated from paleodeposits. We also cannot use our $D_{e,CAM}$ - $D_{e,bootMAM}$ test on silt because each silt aliquot yields an average signal arising from over 1 million grains, meaning there is little to no inter-aliquot scatter, so that the bootMAM and the CAM both give an average dose.

Because we are confident that the bootMAM yielded accurate doses (and ages) for the paired sands (given agreement with stratigraphic constraints and independent age control), we think that testing silt against the sand age obtained with the bootMAM is a reasonable way to check bleaching of the silt.

For this comparison, we present ages rather than doses because the dose rates experienced by sand-sized and silt-sized grains are different, even within the same sediment matrix (due to differences in grain-size dependent attenuation and internal dosing). This means that doses of paired sand and silt are not directly comparable, however, we present the doses in Table S2.

---

## Referee Report (RR1)

**Dear Andreas, Dear Elizabeth, Dear Jakob,**

First, thank you for the carefully revised manuscript. The authors' response is detailed and covers almost all points raised. I am ok with most of the provided answers and changes. However, after carefully reading the provided answers along with the 2nd review and revised manuscript, I realised that the entire manuscript comes with a significant weakness:

In its current shape, the manuscript is somehow extended testing of the method presented in Chamberlain et al. (in press). In one regard this is consequent, and here I do agree with the authors' response, one single paper would have been too long. So this paper is a logical step, yet, you consider your work as "explorative research", without "clear expectations beforehand". It makes it tricky to evaluate the study design. Still without a clear hypothesis to test, the given answers remain ambiguous and the manuscript cannot really decide whether it wants to decipher surface processes or surface dynamics (environmental process focus), or it wants to hand in an extended application test (methodological focus) of the method by Chamberlain et al. (in press).

In their response to the 2nd reviewer, the authors deny the need for a simulation that could test the method by arguing that this would complicate and lengthen the manuscript. Contrary, I believe that a properly developed hypothesis regarding surface dynamics paired with a simulation of the method (that can be compared) would dramatically strengthen the manuscript and make it more rigour.

Similar, the manuscript still starts with "Luminescence dating ... " (before "Optical dating ...)", the interesting part, 'sediment pathways' can be found line 87–94. Given the journal scope, it is starting at the wrong end of the stick.

I did not give much thought beforehand as I suggested to the authors to ship some BIN/BINXfile(s) along with the manuscript allowing readers and reviewers to play with the method parameters. The more surprising was the answer: "Reasons are that the data is not sole property of the authors, and that follow- on publications are planned using part of the same dataset. Publishing the data at this point would jeopardize these publications." I understand that the authors used raw data from published work from five different studies, in three out of this five studies the first author is also the first author, so data property should not be an issue. Besides, I would not expect that the authors reveal their entire dataset (for understandable reasons), but share, for example, one single BIN/BINX-file from one sample of their choice. These data alone is of no value to others, except for verifying the here used method. I cannot see how future publications would become jeopardised. Please consider again to add a file of your choice enabling a proper cross-check of your results.

Since my comments above probably read too negative, I should emphasise (again) that I am in favour of the article and the work done here. Still I think that the manuscript could be stronger and my comments aiming at this goal.

**1 Detailed remarks**

**1.1 Response to authors' comments**

1. I acknowledge that you have reworked the introduction, but the first paragraph feels still misplaced given the target audience of the journal. The primary focus should be the process, and then you highlight the benefits of luminescence dating the method. On the

other hand, I don't want to interfere too much with a generally well-written text. Means, my recommendations still stands, the decision is yours.

2. "It is the standard deviation of the Des divided by the square root of n.":

Please share this information also with the reader by making it part of the main text, i.e. it should read "standard error of the mean" where applicable.

3. "We do not feed a sigma\_b to the CAM. We feed a sigma\_b of [0 0] to the bootMAM, when modeling the overdispersion of the dataset. This is now described in section 3.2.."

In other words, your initial sigma\_b for the CAM is 0. Since you did not share information on the software you used to calculate the CAM (self-written Excel sheet, *MatLab*?), I can only speculate that your CAM calculation does not allow any sigma\_b as input to account for any additional overdispersion that may also be observed for well-bleached samples (e.g., Cunningham & Wallinga, 2012). This is not important, as long you make it clear in the text and add some words on how do you justify this assumption. Your answer should cover the case that people want to start with a (justified and not extreme) pre-assumption on sigma\_b (e.g., based on information on dose rate heterogeneity). If your approach is robust, it should allow such values without altering the overall outcome of your study.

- 4. "We used a conventional excel sheet with standard inputs to determine dose rates; we do not think this detail is needed in the manuscript.".The readers have a right to know how you did calculate your results (you have the supplement if you don't like to have it mentioned in the main text).
- 5. "...calculated by dividing the paleodose of each sample by its dose rate, and propagating uncertainties in quadrature."
  What did you do then with the systematic uncertainties?
- 6. "In our mind, both remnant dose and remnant age are of importance. The remnant dose upon deposition is the most direct measurement of the degree of bleaching, and thus relevant when investigation the dependency of bleaching on depositional context or sample properties. Yet, when dating fluvial samples, one is concerned about potential age overestimation due to incomplete bleaching. Therefore the 'remnant age', as inferred from the remnant dose in combination with the sample dose rate, is also relevant."

I guess your answer means that you did not check for the impact for micro-dosimetry effects, right?

I do agree that it appears more appealing to translate your information into an age, which can be easily understood by readers not familiar with the method. Problematic is that by doing so, you make critical assumptions about the distribution of a variable you did not observe (dose rate). Contrary, looking only at the luminescence, i.e. something you indeed observe and normalise these values to its sensitivity, would give you information on the signal bleaching. The drawback is that you answer will be somehow binary (bleached, not bleached), so talking about a dose and translate this into an age sounds more precise, but your inference becomes much more complex and includes a lot of "ifs".

Nevertheless, I don't want to diminish your work, but you have to make your decision clear in the text to the reader. Means, your discussion and your conclusion should point out, in understandable words for non-specialists, that your data and your approach holds only for the assumptions you made (which should be repeated).

7. "Although we warmly support full access to data, this is unfortunately not possible for these data at this point in time. Reasons are that the data is not sole property of the authors, and that follow- on publications are planned using part of the same dataset. Publishing the data at this point would jeopardize these publications. Yet, the methods are fully documented in our papers, and can be applied to other large datasets. If problems occur, we would be happy to assist and support."

I did not ask for full access to **all** data. What I did ask has no value to others, except for the sole purpose of cross-checking the results and playing with the data. This should be in your interest. Besides, I presume that other people likely want to apply your method in the future. Having a reference would allow them to spot their own mistakes and it makes your paper stronger.

**1.2 Main text**

- 1. Line 202: Change to 'BIN/BINX-files'
- 2. Line 196–198: Please add the grain size you refer to in brackets. "Sand" and "silt" covers a large range, but you use a few different grain ranges, it should be clear all the time to which you are referring to. In particular, sometimes you use "fine silt" instead of 'silt", but given your data, I guess you always refer to  $4-20 \,\mu\text{m}$ .
- 3. Line 249: "Remnant" ¿¿ "remnant"
- 4. Line 250–255: This paragraph presents good reasoning to explain why you prefer the residual dose instead of a residual age. However, micro-dosimetric effects remain still unaccounted.
- 5. Line 274: Replace "estimated" by "approximated" (with this you remain consistent with your arguments given just a few lines above). You should check this throughout and more clear that any "age translation" is highly speculative and not necessarily supported by your data.
- 6. Line 330: "1 to" ¿¿ "1 Gy to"
- 7. Line 366: Too unspecific ("finer sand grains" and "coarser grains"). Please refer to the fractions you investigated.
- 8. Line 371–372: Please supplement this with brief information about the proportion of the sediment load moving in suspension and the sediment moving as bedload (approximated for your sampling). It is important to relate your findings to the actual transport process and its transport energy.
- 9. Line 376: The paper by Fuchs et al. (2005) does not help. Their study features the bleaching differences between quartz and feldspar. I think that this is not the same what you have done here.
- 10. Line 459: Please better refer to 'new' than 'established' here.
- 11. Line 464: You should rephrase the first point, to "Luminescence signal bleaching of sediments can be highly temporally and spatially variable".

esurf-2018-76

- 12. Line 467–468: Please reread these lines: You start with "dating purposes" and then add in brackets "overdispersion', and then you tell the reader, please do not sample sites you are not interested. This reads a little bit odd.
- 13. Line 471–473: In my opinion this a correct observation, but a wrong conclusion. The turbidity brings your particles in suspension and the suspension time determines your bleaching, currently, it reads as if you have a different process that controls your bleaching. Besides, if you bring this in the conclusion, you should also have it as an own paragraph in the discussion before (please see also my comments to the tidal influence below) and it should not come by surprise.
- 14. Line 523: I could not access the PhD thesis "Dating Deltas". I tried to get my hands on it via the usual university library networks (without success), and then I found an abstract in the "Tulane University Digital Library". Probably I wasn't trying hard enough, but maybe you can add a link, DOI or a least an ISBN? This would be somehow important since you refer in the manuscript to it when it comes to the suspended samples.

**1.3 Figures**

1. Figure 1: I've overlooked it the last time, please indicate the boundary of the entire delta. Similar important would be (an approximated) range of the influence of the tide since you argue finally with 'upwelling of turbid water'.

Sebastian Kreutzer — Montréal, 2019–01–30

**References**

Chamberlain, E.L., Wallinga, J., Shen, Z., 2018. Luminescence age modeling of variablybleached sediment: Model selection and input. Radiation Measurements 1–7. doi: 10.1016/j.radmeas.2018.06.00

Cunningham, A.C., Wallinga, J., 2012. Realizing the potential of fluvial archives using robust OSL chronologies 12, 98–106. doi:10.1016/j.quageo.2012.05.007 Fuchs, M., Straub, J., Zöller, L., 2005. Residual luminescence signals of recent river flood sediments: A comparison between quartz and feldspar of fine- and coarse-grain sediments. Ancient TL 23, 25–30.